# An Efficient Dataset Condensation Plugin and Its Application to Continual Learning

**Enneng Yang**[1], **Li Shen**[2*], **Zhenyi Wang**[3*], **Tongliang Liu**[4], **Guibing Guo**[1*]

[1]Northeastern University, China [2]JD Explore Academy, China
[3]University of Maryland, College Park, USA [4]The University of Sydney, Australia
ennengyang@stumail.neu.edu.cn; mathshenli@gmail.com; zwang169@umd.edu
tongliang.liu@sydney.edu.au; guogb@swc.neu.edu.cn

## Abstract

Dataset condensation (DC) distills a large real-world dataset into a small synthetic dataset, with the goal of training a network from scratch on the latter that performs similarly to the former. State-of-the-art (SOTA) DC methods have achieved satisfactory results through techniques such as accuracy, gradient, training trajectory, or distribution matching. However, these works all perform matching in the high-dimension pixel space, ignoring that natural images are usually locally connected and have lower intrinsic dimensions, resulting in low condensation efficiency. In this work, we propose a simple-yet-efficient dataset condensation plugin that matches the raw and synthetic datasets in a low-dimensional manifold. Specifically, our plugin condenses raw images into two low-rank matrices instead of parameterized image matrices. Our plugin can be easily incorporated into existing DC methods, thereby containing richer raw dataset information at limited storage costs to improve the downstream applications' performance. We verify on multiple public datasets that when the proposed plugin is combined with SOTA DC methods, the performance of the network trained on synthetic data is significantly improved compared to traditional DC methods. Moreover, when applying the DC methods as a plugin to continual learning tasks, we observed that our approach effectively mitigates catastrophic forgetting of old tasks under limited memory buffer constraints and avoids the problem of raw data privacy leakage.

## 1 Introduction

Modern machine learning techniques utilize large-scale real-world datasets and advanced deep neural networks (DNNs) to achieve amazing success in various fields, such as models like SAM [24] and GPT [6, 38] in the fields of computer vision and natural language processing, both of which have surpassed classical models [51, 42, 43] trained on small datasets. However, training a well-performing model in the machine learning community requires repeated tuning of various aspects of the model [14], such as the number of layers, learning rate, and other important hyper-parameters. When the dataset is large, the cost of data management and repeated model training becomes unacceptable. As stated in [65], NAS-Bench-101 [63] spent 100 TPU-years of computing time on the CIFAR10 dataset [26] for an exhaustive neural architecture search. In contrast, it only takes dozens of TPU minutes to train the best model from scratch using the optimal configuration discovered [65]. Additionally, in continual learning [41, 2, 60, 61], to prevent forgetting old tasks while learning new tasks, a certain amount of old task data is typically stored for replay [58, 57, 2, 41]. Nevertheless, storing large old task datasets is unrealistic due to strict memory and privacy constraints. Therefore, reducing the data size becomes a valuable and emerging research direction.

---

[*]Corresponding authors.

37th Conference on Neural Information Processing Systems (NeurIPS 2023).

A direct way to reduce data size is to select a representative subset from the original dataset [17]. This paradigm calculates the importance score for each sample based on specific importance criteria and selects a subset to replace the entire training dataset [62]. The calculation criteria include the distance between the sample and the class center [10], the gradient norm of sample [39], and the diversity among samples [47, 57], etc. However, the selection-based method is not always effective, particularly when the task-condition data information is uniformly distributed in the original data [30]. In such cases, the heuristic coreset selection method can only capture a limited amount of information. Recent research [56] has shown that generating a small dataset that performs similarly to the original dataset, instead of selecting a subset, is a promising direction. Therefore, dataset condensation (DC) or dataset distillation (DD) [56, 66, 64, 65, 37, 11, 15] has recently received increasing attention. This novel paradigm distills information from a large real-world dataset into a compact synthetic dataset that produces comparable results to the original dataset. The earliest DD [56] method uses accuracy value as the direct distillation objective and describes the distillation process as a bi-level optimization problem. However, matching accuracy directly through bi-level optimization involves high computational costs and memory overhead [15]. Therefore, state-of-the-art (SOTA) DC methods perform condensation through surrogate objectives, such as gradient [66, 64, 22], training trajectory [8], feature [55], or distribution [65] matching, and achieve more satisfactory results.

However, existing DC methods all optimize parameterized condensed images in the original high-dimensional pixel space, overlooking the fact that natural images typically exhibit local connectivity and have low intrinsic dimensionality [31]. More specifically, in the visual self-supervision task, Masked Autoencoder [18] divides the image into multiple patches and reconstructs the complete image by randomly masking a certain proportion of patches. Experimental results show that masking 75% of the patches can still reconstruct the original image. Also, Maximum Likelihood Estimation [31, 40] of the image's intrinsic dimensions on ImageNet [12] dataset shows that although each image contains $150,528$ pixels, its intrinsic dimension is only between $26$ and $43$. We further verify in the appendix that both the original images and the images generated by traditional dataset distillation (e.g., DSA [64], DM [65]) are low rank. These results imply that compressing data in high-dimensional pixel space is an inefficient approach.

In this work, we propose a simple yet efficient plugin for dataset condensing that compresses a large dataset into a compact synthetic dataset on a low-dimensional manifold. Specifically, unlike existing DC methods that train a compact dataset $\mathcal{S}$ of size $N \times D \times H \times W$, where $N$, $D$, $H$, and $W$ are the number, channels, height, and width of images, respectively, we decompose an image representation under each channel $ch \in \{1, \ldots, D\}$ into a low-dimensional space, and learn two low-rank matrices $\mathcal{A} \in \mathbb{R}^{N \times D \times H \times r}$ and $\mathcal{B} \in \mathbb{R}^{N \times D \times r \times W}$, where $r << \min\{H, W\}$ represents the rank size. Obviously, our low-rank method is orthogonal to existing DC methods and can be integrated into SOTA DC methods as a flexible plugin to improve the learning efficiency of DC. Without loss of generality, we conduct extensive experiments on multiple publicly real-world datasets, integrating the proposed plugin into gradient matching-based [66, 64] and distribution matching-based [65] DC methods. Experimental results show that using our low-rank plugin significantly reduces storage consumption for compact datasets and achieves comparable performance as SOTA DC methods based on high-dimensional pixel spaces, with the same number of images. Furthermore, under the same memory consumption, our plugin can effectively store more information from the large dataset, thus significantly improving the performance of condensed data in downstream tasks. Lastly, when applying low-rank DC methods to continual learning (CL) tasks, we observe that our approach effectively mitigates catastrophic forgetting of old tasks under the constraints of limited memory buffers and avoids data privacy issues by storing condensed rather than raw data.

The main contributions of this paper are as follows:

- We propose a simple yet effective dataset distillation plugin that condenses a large dataset into a compact synthetic dataset on a low-dimensional manifold, offering an orthogonal approach to existing DC methods.
- Experiments on deep learning tasks demonstrate that the proposed plugin achieves performance comparable to that of existing SOTA DC methods while significantly reducing memory consumption. It achieves significantly better accuracy than existing SOTA DC methods with the same memory consumption.
- We also verify that the dataset condensed in the low-dimensional manifold has good cross-architecture transferability and maintains the excellent characteristics of existing DC methods, such as learning the class distribution information of the large dataset.

- When applying low-rank DC as a plugin for CL tasks, our DC plugin approach effectively mitigates the problem of catastrophic forgetting of old tasks while protecting the data privacy.

## 2 Related Works

In this section, we summarize the most related work to this work as three-fold, including coreset sample selection and dataset distillation; continual learning; and low-rank manifolds.

**Coreset Selection and Data Condensation.** Coreset selection [17] and dataset condensation [56] are two methods to eliminate data redundancy, which help to improve the model's training efficiency and reduce the cost of data management. Coreset selection has been widely studied in active learning [48] and continual learning [34], which tries to identify the most informative training samples [62, 59, 10, 44, 3]. Unfortunately, these methods fail when the task-condition information is evenly distributed among the original samples [30]. Empirical studies [56, 66] show that the benefit of existing coreset selection methods is marginal compared with random selection across multiple benchmarks. In recent years, dataset condensation [56] has been proposed to distill a large real dataset onto a small synthetic dataset and achieve better performance than coreset selection. DD [56] formulates the dataset distillation as a bi-level optimization. The inner loop utilizes the condensed dataset to train a network, while the outer loop minimizes the prediction error of the trained network on the original dataset. It updates the synthesized dataset pixel by pixel using Back-Propagation Through Time [35] to compute the meta gradient. KRR [36] transforms the distillation problem into a kernel ridge regression problem, simplifying the expensive nested optimization in DD [56] to a first-order optimization. Unlike the aforementioned works that optimize based on accuracy value, recent works believe that the effectiveness of models trained on the larger dataset and the compact dataset can be reflected in their corresponding parameter states or gradient states [15]. Therefore, they choose to optimize more fine-grained surrogate objectives [66, 64, 30, 8, 65]. Notably, DC [66] and DSA [64] minimize the gradient matching loss between the large dataset and the synthetic dataset at each training step, MTT [8] matches the training trajectory of parameters, LCMat [50] matches loss curvature, CAFE [55] aligns layer-wise features, and DM [65] generates synthetic samples that resemble the distribution of real samples in the feature space. There has recently been a small amount of work on parameter-efficient dataset distillation. For example, IDC [23] and IDM [67] take advantage of the local similarity characteristics of images to partition and expand condensed data to generate more images at the same memory cost. HaBa [33] and RememberThePast [13] utilize bases to produce images or share bases among all classes respectively. However, these methods ignore that natural images have low intrinsic dimensionality [40], which leads to inefficient condensation.

**Continual Learning.** Continual learning (CL) aims to enable a neural network model to learn new tasks continuously without forgetting the old tasks [68, 58]. Inspired by the working mechanism of the human brain, mainstream memory-based CL methods consolidate previously learned knowledge by replaying old data, thereby avoiding catastrophic forgetting. Due to strict memory and privacy constraints, usually only a small portion of old task data can be kept, many above-mentioned coreset selection methods are used to select informative samples for storage. For example, some heuristic method is used to select the most representative samples from each class [59] (e.g., iCaRL [44]) or the sample closest to the decision boundary (e.g., Rwalk [9], MIR [2], Shim [49]). However, the number of new task's samples and stored old samples in memory is often highly unbalanced, leading the model to be biased towards learning new tasks with more data. Recent works [7, 41] addressing this class imbalance in memory-based CL has achieved impressive and satisfactory performance: End-to-End Incremental Learning [7] samples the data after training on the new classes and combines it with the data in memory to create balanced data for fine-tuning. GDumb [41] greedily stores samples in memory and trains the model from scratch on new tasks using only the samples in memory.

**Low-rank Manifolds.** The low-rank structure is prevalent in machine learning, such as computer vision [40, 18, 32], natural language processing [20, 1], recommendation systems [25, 45], etc. For example, MLE [31, 40] found that natural image datasets containing thousands of pixels can actually be described with fewer variables. LoRA [20] can fine-tune the GPT-3 175B [6] with comparable accuracy to full-parameter fine-tuning while reducing the number of trainable parameters by a factor of 10,000 and GPU memory requirements by a factor of three. In recommender systems, the original highly sparse rating/interaction matrix is usually decomposed into low-rank user embedding matrices and item embedding matrices [25] to capture user interests and item features. However, to the best

of our knowledge, no research has explored the exploitation of low-rank structure to improve the condensation efficiency for the dataset distillation task, and this work is the first to fill this gap.

## 3 Low-Rank Data Condensation Plugin

In this section, we first define the problem of DC in Sec. 3.1, then introduce our proposed low-rank plugin for DC in Sec. 3.2, and then describe how to integrate our plugin with existing DC methods in Sec. 3.3 and further apply it to continual learning in Sec. 3.4.

### 3.1 Problem Definition

The goal of dataset condensation (DC) is to distills a large target dataset $\mathcal{T} = \{\mathbf{x}_i, y_i\}_{i=1}^{N_{\mathcal{T}}}$ containing $N_{\mathcal{T}}$ training image $\mathbf{x}_i \in \mathbb{R}^d$ and its label $y_i \in \{1, 2, \ldots, |Y|\}$ into a small dataset $\mathcal{S} = \{\mathbf{s}_i, y_i\}_{i=1}^{N_{\mathcal{S}}}$ with $|\mathcal{S}|$ synthetic image $\mathbf{s}_i \in \mathbb{R}^d$, where $N_{\mathcal{S}} \ll N_{\mathcal{T}}$ ($2-3$ orders of magnitude), $|Y|$ represents the number of classes, and $\mathbb{R}^d$ defines a $d$-dimensional space. We expect a network $\phi_{\theta^{\mathcal{S}}}$ trained on the small dataset $\mathcal{S}$ to have similar performance to a network $\phi_{\theta^{\mathcal{T}}}$ trained on the large training set $\mathcal{T}$ on the unseen test dataset, that is:

$$\mathbb{E}_{\mathbf{x}_i \sim P_{\mathcal{T}}} \left[ \ell \left( \phi_{\theta^{\mathcal{T}}}(\mathbf{x}_i), y \right) \right] \simeq \mathbb{E}_{\mathbf{x}_i \sim P_{\mathcal{T}}} \left[ \ell \left( \phi_{\theta^{\mathcal{S}}}(\mathbf{x}_i), y \right) \right],$$

$$\text{s.t. } \theta^{\mathcal{T}} = \operatorname*{arg\,min}_{\theta^{\mathcal{T}}} \mathcal{L}^{\mathcal{T}}(\theta^{\mathcal{T}}) = \operatorname*{arg\,min}_{\theta^{\mathcal{T}}} \frac{1}{N_{\mathcal{T}}} \sum_{(\mathbf{x}_i, y) \in \mathcal{T}} \ell \left( \phi_{\theta^{\mathcal{T}}}(\mathbf{x}_i), y \right), \quad (1)$$

$$\theta^{\mathcal{S}} = \operatorname*{arg\,min}_{\theta^{\mathcal{S}}} \mathcal{L}^{\mathcal{S}}(\theta^{\mathcal{S}}) = \operatorname*{arg\,min}_{\theta^{\mathcal{S}}} \frac{1}{N_{\mathcal{S}}} \sum_{(\mathbf{x}_i, y) \in \mathcal{S}} \ell \left( \phi_{\theta^{\mathcal{S}}}(\mathbf{x}_i), y \right),$$

where $P_{\mathcal{T}}$ represents the real distribution of the test dataset, $\mathbf{x}_i$ represents the input image, $y$ represents the ground truth, and $\ell(\cdot)$ represents a loss function such as cross-entropy loss.

To achieve the goal stated in Eq. 1, existing DC methods [56, 66, 64, 8, 65] first initialize the dataset $\mathcal{S} \in \mathbb{R}^{N_{\mathcal{S}} \times D \times H \times W}$ as a set of learnable parameters in high-dimensional pixel space. Here, $N_{\mathcal{S}}$ denotes the number of synthetic images, and $C$, $H$, and $W$ represents the number of channels, the image's height, and the image's width, respectively. Then, the dataset $\mathcal{S}$ is updated pixel-by-pixel based on accuracy value matching [56, 36] or surrogate objective matching [66, 8, 65] of the condensed dataset $\mathcal{S}$ and the large dataset $\mathcal{T}$. For example, in the first dataset distillation work DD [56], dataset $\mathcal{S}$ is treated as a hyperparameter in a bi-level optimization problem as follows:

$$\mathcal{S}^* = \operatorname*{arg\,min}_{\mathcal{S}} \mathcal{L}^{\mathcal{T}} \left( \phi_{\theta^{\mathcal{S}}} \right), \text{ subject to } \theta^{\mathcal{S}} = \operatorname*{arg\,min}_{\theta} \mathcal{L}^{\mathcal{S}}(\phi_{\theta}), \quad (2)$$

where the inner loop trains a randomly initialized network $\phi$ (parameters denoted as $\theta$) on the synthetic dataset $\mathcal{S}$ until convergence, and the outer loop uses the large target dataset $\mathcal{T}$ as a validation set to optimize $\mathcal{S}$. The small dataset $\mathcal{S}$ is updated by solving the meta-gradient [35] of the bi-level optimization in Eq. 2, allowing the trained model $\phi$ on the dataset $\mathcal{S}$ to perform well on the real dataset $\mathcal{T}$. Instead of optimizing directly based on the accuracy value of the distilled data $\mathcal{S}$, SOTA DC methods are based on surrogate objectives to make the model trained on $\mathcal{S}$ and $\mathcal{T}$ approximate each other in the parameter space [66], i.e., $\theta^{\mathcal{T}} \simeq \theta^{\mathcal{S}}$, or in the gradient space [64, 66], i.e., $\nabla_{\theta} \mathcal{L}^{\mathcal{T}}(\theta) \simeq \nabla_{\theta} \mathcal{L}^{\mathcal{S}}(\theta)$, or in the feature distribution space [65], i.e., $\phi_{\theta}(\mathbf{x}_i) \simeq \phi_{\theta}(\mathbf{s}_i)$. However, these methods all focus on pixel-by-pixel optimizations of $\mathcal{S}$, ignoring that natural images are locally connected [18] and usually have low intrinsic dimensionality [32, 40]. In the appendix, we perform principal component analysis on real images and images obtained by traditional dataset distillation methods (such as DSA [64] and DM [65]) and show that they have low-rank properties. Hence, optimizing $\mathcal{S}$ in a high-dimensional pixel space is inefficient.

### 3.2 Our Low-Rank Data Condensation Plugin

In this work, we introduce a low-rank dataset condensation plugin that distills knowledge from a large dataset $\mathcal{T}$ to a small synthetic dataset $\mathcal{S}$ in a low-rank manifold. Specifically, instead of directly initializing $\mathcal{S}$ as a variable with shape $N_{\mathcal{S}} \times D \times H \times W$, we conduct a low-rank decomposition of the content $\mathbf{x}_{i,ch} \in \mathbb{R}^{H \times W}$ in the channel $ch \in \{1, \ldots, D\}$ of an image $\mathbf{x}_i$ and employ two variables, $\mathcal{A}_{i,ch} \in \mathbb{R}^{H \times r}$ and $\mathcal{B}_{i,ch} \in \mathbb{R}^{r \times W}$, to reconstruct a high-dimensional image by $\mathbf{x}_{i,ch} \approx \mathcal{A}_{i,ch} \mathcal{B}_{i,ch}$,

**Algorithm 1:** LoDC: Low-rank Dataset Condensation with Gradient Matching [66]

**Input:** A large training dataset $\mathcal{T}$

1 **Required**: Randomly initialized $\mathcal{A}, \mathcal{B}$ of rank $r$ for $|Y|$ classes, the probability distribution of randomly initialized weights $P_{\theta_0}$, neural network $\phi_\theta$, number of outer-iterations $T_{ou}$ and number of inner-iterations $T_{in}$, learning rates $\eta_{dc}$ and $\eta_\theta$, minibatch size $N_\mathcal{T}, N_{\mathcal{AB}}$

2 **for** *outer-iteration* $k = 1, \cdots, T_{ou}$ **do**

3     Initialize $\theta_0 \sim P_{\theta_0}$

4     **for** *inner-iteration* $t = 1, \cdots, T_{in}$ **do**

5        **for** *class* $c = 1, \cdots, |Y|$ **do**

6           Sample a minibatch real pair $\mathbf{B}_c^\mathcal{T} \sim \mathcal{T}$ and a minibatch synthetic pair $\mathbf{B}_c^{\Omega(\mathcal{A},\mathcal{B})} \sim \Omega(\mathcal{A}, \mathcal{B})$

7           Compute $\mathcal{L}_c^\mathcal{T}(\theta_t | \mathbf{B}_c^\mathcal{T}) = \frac{1}{N_\mathcal{T}} \sum_{(\mathbf{x}_i, y) \in \mathbf{B}_c^\mathcal{T}} \ell(\phi_{\theta_t}(\mathbf{x}_i), y)$

8           Compute $\mathcal{L}_c^{\Omega(\mathcal{A},\mathcal{B})}(\theta_t | \mathbf{B}_c^{\Omega(\mathcal{A},\mathcal{B})}) = \frac{1}{N_{\mathcal{AB}}} \sum_{(\mathcal{A}_i \mathcal{B}_i, y) \in \mathbf{B}_c^{\Omega(\mathcal{A},\mathcal{B})}} \ell(\phi_\theta(\mathcal{A}_i \mathcal{B}_i), y)$

9           Compute $\mathcal{L}_c^{DC}(\mathbf{B}_c^\mathcal{T}, \mathbf{B}_c^{\Omega(\mathcal{A},\mathcal{B})}) = d\left(\nabla_\theta \mathcal{L}_c^\mathcal{T}(\theta_t | \mathbf{B}_c^\mathcal{T}), \nabla_\theta \mathcal{L}_c^{\Omega(\mathcal{A},\mathcal{B})}(\theta_t | \mathbf{B}_c^{\Omega(\mathcal{A},\mathcal{B})})\right)$

10           Update $\mathcal{A} \leftarrow \mathcal{A} - \eta_{dc} \nabla_\mathcal{A} \mathcal{L}_c^{DC}(\mathbf{B}_c^\mathcal{T}, \mathbf{B}_c^{\Omega(\mathcal{A},\mathcal{B})})$ and $\mathcal{B} \leftarrow \mathcal{B} - \eta_{dc} \nabla_\mathcal{B} \mathcal{L}_c^{DC}(\mathbf{B}_c^\mathcal{T}, \mathbf{B}_c^{\Omega(\mathcal{A},\mathcal{B})})$

11        Update $\theta_{t+1} \leftarrow \theta_t - \eta_\theta \nabla_\theta \mathcal{L}^{\Omega(\mathcal{A},\mathcal{B})}(\theta_t)$

**Output:** A small dataset $\Omega(\mathcal{A}, \mathcal{B})$

where the rank $r \ll \{H, W\}$. Therefore, the goal of data condensation in the low-rank manifold is to optimize $\mathcal{A} \in \mathbb{R}^{N_\mathcal{S} \times D \times H \times r}$ and $\mathcal{B} \in \mathbb{R}^{N_\mathcal{S} \times D \times r \times W}$ such that the network $\phi_{\theta^{\Omega(\mathcal{A},\mathcal{B})}}$, trained on the small reconstructed data $\Omega(\mathcal{A}, \mathcal{B})$, achieves similar performance to the network $\phi_{\theta^\mathcal{T}}$ trained on the high-dimensional large dataset $\mathcal{T}$. Therefore, the DC in the low-rank manifold is formalized as:

$$\mathbb{E}_{\mathbf{x}_i \sim P_\mathcal{T}} [\ell(\phi_{\theta^\mathcal{T}}(\mathbf{x}_i), y)] \simeq \mathbb{E}_{\mathbf{x}_i \sim P_\mathcal{T}} [\ell(\phi_{\theta^{\Omega(\mathcal{A},\mathcal{B})}}(\mathbf{x}_i), y)],$$

$$\text{s.t. } \theta^\mathcal{T} = \arg\min_{\theta^\mathcal{T}} \mathcal{L}^\mathcal{T}(\theta^\mathcal{T}) = \arg\min_{\theta^\mathcal{T}} \frac{1}{N_\mathcal{T}} \sum_{(\mathbf{x}_i, y) \in \mathcal{T}} \ell(\phi_{\theta^\mathcal{T}}(\mathbf{x}_i), y), \tag{3}$$

$$\theta^{\Omega(\mathcal{A},\mathcal{B})} = \arg\min_{\theta^{\Omega(\mathcal{A},\mathcal{B})}} \mathcal{L}^{\Omega(\mathcal{A},\mathcal{B})}(\theta^{\Omega(\mathcal{A},\mathcal{B})}) = \arg\min_{\theta^{\Omega(\mathcal{A},\mathcal{B})}} \frac{1}{N_\mathcal{S}} \sum_{(\mathcal{A}_i \mathcal{B}_i, y) \in \Omega(\mathcal{A},\mathcal{B})} \ell(\phi_{\theta^{\Omega(\mathcal{A},\mathcal{B})}}(\mathcal{A}_i \mathcal{B}_i), y),$$

where $\Omega(\mathcal{A}, \mathcal{B})$ is an operation that reconstructs $\mathbf{x}_i$ channel-wise from low-rank matrices $\mathcal{A}$ and $\mathcal{B}$, i.e., $\mathbf{x}_i = \mathcal{A}_i \mathcal{B}_i = [\mathcal{A}_{i,1} \mathcal{B}_{i,1} | \ldots | \mathcal{A}_{i,D} \mathcal{B}_{i,D}] \in \mathbb{R}^{D \times H \times W}$, where $[\cdot|\cdot]$ means channel-wise stacked the image representation.

**Discussion**. Our method effectively takes advantage of the low intrinsic dimension of natural images, and has the following advantages: (i) The proposed low-rank DC plugin significantly improves condensing efficiency (i.e., fewer training parameters) and reduces the cost of synthetic dataset storage. This is particularly beneficial when the image height ($H$) and width ($W$) are very large. Due to the low intrinsic dimension of the image, the value of $r$ can be very small, such as 2. (ii) Under the same memory consumption or the number of learnable parameters as traditional DC methods, our plugin can reconstruct a larger number of synthetic images, preserving sufficient information for the large training dataset $\mathcal{T}$. (iii) Our plugin is orthogonal to existing dataset distillation work and can be further combined with them.

### 3.3 Incorporating Low-rank DC Plugin to SOTA Methods

Our proposed low-rank manifolds DC plugin can be easily incorporated into existing DC solutions [56, 66, 8, 64, 65]. Without loss of generality, we integrate it into gradient matching-based DC [66] and distribution matching-based DM [65], and define the two new methods as **Lo**w-rank **DC** (LoDC) and **Lo**w-rank **DM** (LoDM), respectively. We verified the effectiveness of the low-rank plugin in Sec. 4.1. Additionally, in the appendix, we combine our low-rank DC plugin with more advanced dataset distillation methods, including MTT [8], IDC [23], HaBa [33], and RemenberThePast [13].

**Low-rank DC (LoDC)**. To achieve the goal stated in Eq. 3, we match the gradients of the large dataset $\mathcal{T}$ and the small dataset $\Omega(\mathcal{A}, \mathcal{B})$ within a low-rank manifold. Specifically, we use the synthetic dataset $\Omega(\mathcal{A}, \mathcal{B})$ to train a deep neural network $\phi_\theta$ (the initial parameter is $\theta_0$). After optimizing the network, we aim to minimize the distance between the gradient $\nabla_\theta \mathcal{L}^\mathcal{T}(\theta_t)$ of the loss $\mathcal{L}^\mathcal{T}$ w.r.t $\theta_t$ on

**Algorithm 2:** LoDM: Low-rank Dataset Condensation with Distribution Matching [65]

---

**Input:** A large training dataset $\mathcal{T}$

1   **Required**: Randomly initialized $\mathcal{A}, \mathcal{B}$ of rank $r$ for $|Y|$ classes, the probability distribution of randomly initialized weights $P_{\theta_0}$, neural network $\psi_\theta$, number of iterations $T_{ou}$, learning rate $\eta_{dc}$.

2   **for** *iteration* $k = 1, \cdots, T_{ou}$ **do**

3       Initialize $\theta_0 \sim P_{\theta_0}$

4       Initialize $\mathcal{L}_s^{DM} = 0$

5       **for** *class* $c = 1, \cdots, |Y|$ **do**

6            Sample a minibatch real pair $\mathbf{B}_c^{\mathcal{T}} \sim \mathcal{T}$ and a minibatch synthetic pair $\mathbf{B}_c^{\Omega(\mathcal{A},\mathcal{B})} \sim \Omega(\mathcal{A}, \mathcal{B})$

7            Compute $\mathcal{L}_c^{DM}(\mathbf{B}_c^{\mathcal{T}}, \mathbf{B}_c^{\Omega(\mathcal{A},\mathcal{B})}) =$
              $d\left(\frac{1}{N_{\mathcal{T}}} \sum_{(\mathbf{x}_i,y) \in \mathbf{B}_c^{\mathcal{T}}} \psi_{\theta_0}(\boldsymbol{x}_i), \frac{1}{N_{\mathcal{AB}}} \sum_{(\mathcal{A}_i\mathcal{B}_i,y) \in \mathbf{B}_c^{\Omega(\mathcal{A},\mathcal{B})}} \psi_{\theta_0}(\mathcal{A}_i\mathcal{B}_i)\right)$

8            Compute $\mathcal{L}_s^{DM} = \mathcal{L}_s^{DM} + \mathcal{L}_c^{DM}(\mathbf{B}_c^{\mathcal{T}}, \mathbf{B}_c^{\Omega(\mathcal{A},\mathcal{B})})$

9       Update $\mathcal{A} \leftarrow \mathcal{A} - \eta_{dc}\nabla_{\mathcal{A}}\mathcal{L}_s^{DM}$ and $\mathcal{B} \leftarrow \mathcal{B} - \eta_{dc}\nabla_{\mathcal{B}}\mathcal{L}_s^{DM}$

**Output:** A small dataset $\Omega(\mathcal{A}, \mathcal{B})$

---

the large dataset $\mathcal{T}$ and the gradient $\nabla_\theta \mathcal{L}^{\Omega(\mathcal{A},\mathcal{B})}(\theta_t)$ of the loss $\mathcal{L}^{\Omega(\mathcal{A},\mathcal{B})}$ w.r.t $\theta_t$ on the small dataset $\Omega(\mathcal{A}, \mathcal{B})$ at step $t$ as DC [66]. That is, we solve the following optimization problem:

$$\min_{\mathcal{A},\mathcal{B}} \mathrm{E}_{\theta_0 \sim P_{\theta_0}} \left[ \sum_{t=1}^{T_{in}} d\left( \nabla_\theta \mathcal{L}^{\mathcal{T}}(\theta_t|\mathcal{T}), \nabla_\theta \mathcal{L}^{\Omega(\mathcal{A},\mathcal{B})}(\theta_t|\Omega(\mathcal{A},\mathcal{B})) \right) \right], \tag{4}$$

where $d(\cdot, \cdot)$ is a distance function, $T_{in}$ is the number of training iterations. $\mathcal{A}$ and $\mathcal{B}$ are updated using gradients obtained by backpropagation in Eq. 4, and $\theta$ is trained using the whole small dataset $\Omega(\mathcal{A}, \mathcal{B})$ by gradient descent optimization. We provide the pseudocode of our LoDC in Algorithm 1. Additionally, LoDC can be easily extended to DSA [64] methods that incorporate differentiable data augmentations into gradient matching [66].

**Low-rank DM (LoDM)**. To achieve the goal of Eq. 3, our LoDM requires the distribution of the small dataset $\Omega(\mathcal{A}, \mathcal{B})$ to accurately approximate the distribution of the real training large dataset $\mathcal{T}$ as DM [65]. Specifically, we first transform each input image $\mathcal{A}_i\mathcal{B}_i \in \mathbb{R}^{(C \times H \times W)}$ into a different space through a family of parametric functions $\psi_\theta: \mathbb{R}^{(C \times H \times W)} \to \mathbb{R}^{d'}$. We then use the maximum mean difference (MMD) [16] to estimate the distance between the real and compact data distributions: $\sup_{\|\psi_\theta\|_{\mathcal{H}} \leq 1} (\mathbb{E}[\psi_\theta(\mathcal{T})] - \mathbb{E}[\psi_\theta(\Omega(\mathcal{A},\mathcal{B}))])$, where $\mathcal{H}$ is reproducing kernel Hilbert space. As the real data distribution is unavailable, we utilize the empirical estimate of MMD as DM [65], that is, LoDM to solve the following optimization problem:

$$\min_{\mathcal{A},\mathcal{B}} \mathbb{E}_{\theta_0 \sim P_{\theta_0}} \left[ d\left( \frac{1}{N_{\mathcal{T}}} \sum_{i=1}^{N_{\mathcal{T}}} \psi_{\theta_0}(\boldsymbol{x}_i), \frac{1}{N_{\mathcal{AB}}} \sum_{i=1}^{N_{\mathcal{AB}}} \psi_{\theta_0}(\mathcal{A}_i\mathcal{B}_i) \right) \right], \tag{5}$$

where $N_{\mathcal{T}}$ and $N_{\mathcal{AB}}$ represent the sample sizes of $\mathcal{T}$ and $\Omega(\mathcal{A}, \mathcal{B})$ respectively. The low-rank matrices $\mathcal{A}$ and $\mathcal{B}$ are updated by performing a gradient descent on Eq. 5. DM [65] indicates that network $\psi_{\theta_0}$ can perform well when a family of random initializations is adopted. Therefore, $\theta_0$ does not need to be updated but is randomly sampled in each training iteration. We have adopted this default setting in this paper. Algorithm 2 provides the optimization process of our LoDM.

### 3.4 Application to Continual Learning

The goal of class continual learning (CL) is to use a model to learn from a continuously arriving sequence of new classes while retaining knowledge from previous classes [68, 58]. Mainstream memory-based CL methods typically involve carefully selecting a small number of old samples from a large training dataset and storing them in a limited memory buffer $\mathcal{M}$ to mitigate catastrophic forgetting [59, 7, 41, 57]. The degree of forgetting of old classes is directly affected by the amount of data information in the samples stored in the limited memory buffer $\mathcal{M}$. Our dataset condensation plugin can compress a raw large dataset into a smaller dataset $\Omega(\mathcal{A}, \mathcal{B})$ by Alg. 1 or Alg. 2 to story in $\mathcal{M}$, thereby to preserve more information about older classes under strict memory constraints. In this work, similar to DM [65], we establish a baseline using the simple yet effective GDumb [41] approach

Table 1: Comparison with coreset selection methods and dataset condensation methods.

| DataSet | Img/Cls | Ratio% | Coreset Selection Methods | | | Dataset Condensation Methods | | | | | |
|---|---|---|---|---|---|---|---|---|---|---|---|
| | | | Random | Herding | Forgetting | DD | LD | DC | DSA | DM | LoDM(Ours) |
| MNIST | 1 | 0.017 | 64.9±3.5 | 89.2±1.6 | 35.5±5.6 | - | 60.9±3.2 | **91.7±0.5** | 88.7±0.6 | 89.7±0.6 | 91.2±0.4 |
| | 10 | 0.17 | 95.1±0.9 | 93.7±0.3 | 68.1±3.3 | 79.5±8.1 | 87.3±0.7 | 97.4±0.2 | 97.1±0.1 | 96.5±0.2 | **97.7±0.1** |
| | 50 | 0.83 | 97.9±0.2 | 94.8±0.2 | 88.2±1.2 | - | 93.3±0.3 | 98.8±0.2 | **99.2±0.1** | 97.5±0.5 | 98.2±0.1 |
| CIFAR10 | 1 | 0.02 | 14.4±2.0 | 21.5±1.2 | 13.5±1.2 | - | 25.7±0.7 | 28.3±0.5 | 28.8±0.7 | 26.0±0.8 | **43.8±0.8** |
| | 10 | 0.2 | 26.0±1.2 | 31.6±0.7 | 23.3±1.0 | 36.8±1.2 | 38.3±0.4 | 44.9±0.5 | 51.1±0.5 | 48.9±0.6 | **59.8±0.4** |
| | 50 | 1 | 43.4±1.0 | 40.4±0.6 | 23.3±1.1 | - | 42.5±0.4 | 53.9±0.5 | 60.6±0.5 | 63.0±0.4 | **64.6±0.1** |
| CIFAR100 | 1 | 0.2 | 4.2±0.3 | 8.4±0.3 | 4.5±0.2 | - | 11.5±0.4 | 12.8±0.3 | 13.9±0.3 | 11.4±0.3 | **25.6±0.5** |
| | 10 | 2 | 14.6±0.5 | 17.3±0.3 | 15.1±0.3 | - | - | 25.2±0.3 | 32.3±0.3 | 29.7±0.3 | **37.5±0.8** |
| TinyImageNet | 1 | 0.2 | 1.4±0.1 | 2.8±0.2 | 1.6±0.1 | - | - | 4.61±0.2 | 4.79±0.2 | 3.9±0.2 | **10.3±0.2** |
| | 10 | 2 | 5.0±0.2 | 6.3±0.2 | 5.1±0.2 | - | - | 11.6±0.3 | 14.7±0.2 | 12.9±0.4 | **18.3±0.3** |

Table 2: Comparison with dataset distillation methods on the same image (SI) or same memory (SM).

| DataSet | Img/Cls | DC | LoDC(SI) | LoDC(SM) | DSA | LoDSA(SI) | LoDSA(SM) | DM | LoDM(SI) | LoDM(SM) |
|---|---|---|---|---|---|---|---|---|---|---|
| MNIST | 1 | 91.7±0.5 | - | **93.0±0.3** | 88.7±0.6 | - | **90.6±0.6** | 89.7±0.6 | 87.0±0.7 | **91.2±0.4** |
| | 10 | 97.4±0.2 | 96.0±0.2 | **97.6±0.3** | 97.1±0.1 | 95.3±0.2 | **97.7±0.1** | 96.5±0.2 | 92.0±0.6 | **97.7±0.1** |
| CIFAR10 | 1 | 28.3±0.5 | 28.2±0.5 | **35.2±0.5** | 28.8±0.7 | 28.3±0.6 | **41.0±0.2** | 26.0±0.8 | 24.8±0.3 | **43.8±0.8** |
| | 10 | 44.9±0.5 | 42.8±0.4 | **50.8±0.3** | 51.1±0.5 | 47.6±0.4 | **56.5±0.3** | 48.9±0.6 | 46.0±0.7 | **59.8±0.4** |
| CIFAR100 | 1 | 12.8±0.3 | 12.3±0.2 | **18.6±0.1** | 13.9±0.3 | 13.5±0.2 | **22.9±0.2** | 11.4±0.3 | 9.6±0.4 | **25.6±0.5** |
| | 10 | 25.2±0.3 | 24.4±0.2 | **27.1±0.6** | 32.3±0.3 | 29.8±0.1 | **33.7±0.7** | 29.7±0.3 | 27.7±0.2 | **37.5±0.8** |
| TinyImageNet | 1 | 4.61±0.2 | 4.21±0.2 | **6.80±0.1** | 4.79±0.2 | 4.70±0.3 | **9.90±0.4** | 3.9±0.2 | 3.6±0.1 | **10.3±0.2** |

in CL. GDumb first stores training samples in memory in a greedy manner while ensuring balanced samples per class. During testing, the model is trained from scratch exclusively using samples from memory $\mathcal{M}$. More specifically, we compare the effects of storing randomly selected sample sets [41], Herding sample sets [59, 44, 7], traditional DC condensed dataset (i.e., DC [66]/DSA [64], DM [65]), and our low-rank manifold condensed dataset (i.e., LoDC or LoDM in Sec. 3.3) in memory $\mathcal{M}$ for CL in Sec. 4.2.

# 4 Experiment

In this section, we conduct experiments to verify the effectiveness of the proposed low-rank DC plugin. Due to space constraints, some experiment results are included in the **Appendix**.

## 4.1 Data Condensation for Deep Learning

**Datasets**. We evaluate our low-rank DC plugin on four benchmark datasets as DM [65], including MNIST [29], CIFAR10 [26], CIFAR100 [26], and TinyImageNet [28]. *MNIST* contains 60,000 grayscale images of size $28 \times 28$, totaling 10 classes. *CIFAR10* and *CIFAR100* contain 50,000 RGB images of size $32 \times 32$. The former has 10 classes, and the latter has 100 classes. *TinyImageNet* contains 100,000 images resized to $64 \times 64$, with a total of 200 classes.

**Baselines**. We compare two kinds of methods to reduce the amount of data: coreset selection (i.e., Random, Herding [59, 4], Forgetting [53]) and synthetic dataset distillation (i.e., DD [56], LD [5], DC [66], DSA [64], DM [65]). Specifically, *Random* sampling selects images randomly, *Herding* heuristically selects the sample closest to the center of the class, and *Forgetting* selects the sample that is most likely to be forgotten during model training. *DD* is one of the earliest works on dataset distillation, which updates the synthetic dataset by solving a bi-level optimization problem. *LD* performs label distillation rather than image distillation. *DC* performs gradient matching between large training dataset and condensed dataset, and *DSA* further considers differentiable siamese augmentation strategies based on DC. *DM* condenses datasets through distribution matching. We further combine the proposed low-rank plugin into DC, DSA and DM to obtain *LoDC*, *LoDSA* and *LoDM*, respectively. Comparisons with other dataset distillation methods (e.g., MTT [8], IDC [23], HaBa [33], and RemenberThePast [13]) are provided in the appendix.

**Experimental Details**. In each experiment, we first choose the coreset (for Random, Herding, Forgetting) or learn a synthetic dataset (for DD, LD, DC, DSA, DM, LoDC, LoDSA, LoDM) and then employ it to train 20 randomly initialized networks (default ConvNet [46] architecture). By default, we set the rank $r$ of our low-rank plugin to 2 for MNIST, CIFAR10, CIFAR100, and 4 for TinyImageNet. In a few cases, the rank $r$ will be searched in $\{1, 2, 4, 8\}$. Other hyperparameters are the same as baselines [66, 64, 65]. We repeat the experiment multiple times for each method and

Table 3: Cross-architecture testing performance on CIFAR10 (10 images per class). *LearnOn* means condensing the dataset on this architecture, and *TestOn* means using the condensed dataset to train a new architecture.

| Method | LearnOn \ TestOn | ConvNet | LeNet | AlexNet | VGG11 | ResNet18 |
|---|---|---|---|---|---|---|
| DSA | AlexNet | 30.4±0.7 | 24.2±0.4 | 28.3±0.4 | 27.2±1.0 | 27.8±1.1 |
| | ConvNet | 31.4±1.1 | 21.7±1.6 | 25.9±0.8 | 27.6±0.8 | 27.6±1.4 |
| DM | AlexNet | 41.4±0.8 | 31.4±0.2 | 37.5±0.9 | 36.8±0.5 | 34.9±1.1 |
| | ConvNet | 42.2±0.5 | 33.4±0.6 | 38.8±1.3 | 36.2±1.0 | 34.6±0.5 |
| LoDM(ours) | AlexNet | 56.2±0.3 | 32.9±0.9 | 49.9±0.5 | 51.0±0.6 | 50.9±0.5 |
| | ConvNet | 56.4±0.3 | 45.5±0.6 | 53.4±0.6 | 50.7±0.6 | 50.6±0.7 |

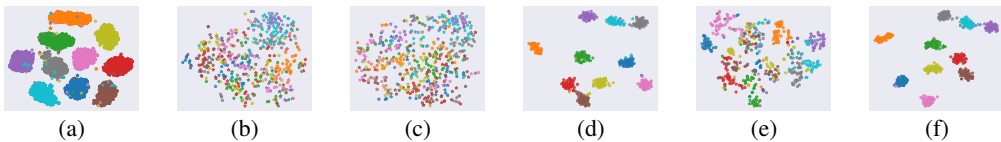

| (a) | (b) | (c) | (d) | (e) | (f) |

Figure 1: Distribution of real (all images) and synthetic images (50 images per class) on MNIST dataset: (a) Real, (b) DC, (c) Our LoDC($r=8$), (d) DM, (e) Our LoDM($r=2$), and (f) Our LoDM($r=8$).

report the average test accuracy. Additionally, in Sec. 4.1.2, we also tested the cross-architecture effectiveness of the synthetic dataset on five standard deep network architectures: ConvNet [46], LeNet [29], AlexNet [27], VGG11 [52], and ResNet18 [19].

### 4.1.1 Compared to SOTA DC Baselines

We compare the performance of our LoDC, LoDSA, LoDM with other baseline methods under different sample sizes (1/10/50 image(s) per class) in Tab. 1 and Tab. 2. We have the following observations from Tab. 1: (i) When the sample size is small (e.g., 1 image per class), both Random and heuristic Forgetting sample selection perform poorly, significantly lower than Herding's method, because the latter samples best represent the class centers. As the sample size of each class increases, Herding's advantage becomes less obvious. (ii) The dataset condensation methods are significantly better than the coreset methods. For example, in CIFAR10 (10 images per class), Random and Herding only achieve 26.0% and 31.6% accuracy, while DD achieves 36.8% accuracy, DC, DM, and DSA achieve 44.9%, 48.9%, and 51.1%, respectively. (iii) Our method condenses dataset in a low-rank manifold, effectively reducing memory cost (or the number of parameters to be optimized per image). Therefore, by utilizing the same memory, our low-rank LoDM can represent a more significant number of images, which is significantly better than other SOTA dataset compression methods, especially when the sample size of each class is small. For example, when using one image per class, our LoDM achieved accuracies of 43.8%, 25.6%, and 10.3% on CIFAR10, CIFAR100, and TinyImageNet, respectively, while the best baseline achieved only 28.8%, 13.9%, and 3.9%.

In Tab. 2, we compare the performance of our methods (LoDC, LoDSA, LoDM) in the low-rank manifold with traditional dataset distillation methods (DC, DSA, DM) using the same number of images (SI) or the same memory consumption (SM). It can be observed that: (i) Our method (rank $r = 2$) reduces storage consumption by $7\times$, $8\times$, and $8\times$ under MNIST, CIFAR10, CIFAR100 datasets, respectively, when the number of images is the same. However, the performance is still relatively close to the traditional DC method. We further explain the correspondence between rank size and memory cost in the appendix. (ii) Under the same memory consumption, our method can condense the large dataset into a smaller dataset with more samples, thereby preserving more information from the original large datasets. Therefore, compared to traditional condensing methods, our method exhibits a significant improvement. For example, on the CIFAR10 (10 Img/Cls) dataset, LoDSA and LoDM have shown an improvement of 8.9% and 13.8% compared to DSA and DM, respectively. Furthermore, as shown in Fig. 2(a), our LoDM method can achieve a higher final performance than DM after only 2,000 iterations, compared to 20,000 iterations.

### 4.1.2 Ablation Study

**Cross-architecture Transferability Analysis**. We verify that the condensed dataset learned in the low-rank manifold still has good cross-architecture transfer ability. We learn condensed dataset on a particular architecture (i.e., AlexNet/ConvNet), and then use the learned small dataset to train five different architectures (use Batch Normalization [21] as DM) from scratch, and finally verify on the test set of CIFAR10 dataset. As shown in Tab. 3, we observe that in each method (DSA, DM or

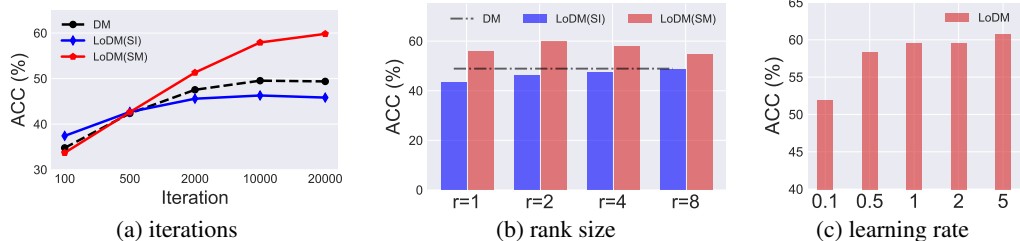

(a) iterations          (b) rank size          (c) learning rate

Figure 2: Ablation study on CIFAR10 (10 images per class): (a) Accuracy changes during DM, LoDM(SI) and LoDM(SM) iterations, (b) rank size of synthetic datasets, and (c) learning rate of synthetic datasets ($r = 2$).

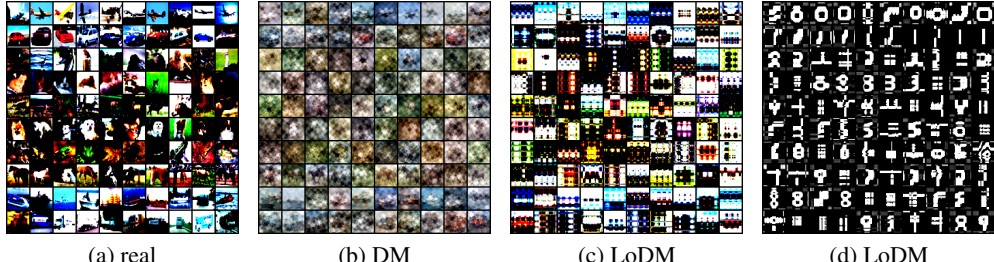

(a) real          (b) DM          (c) LoDM          (d) LoDM

Figure 3: Visualization of (a) real and (b) DM synthetic, and (c) LoDM(Ours) synthetic images on CIFAR10 dataset (10 images per class) and (d) LoDM(Ours) on MNIST dataset (10 images per class).

LoDM), the condensed data on AlexNet and ConvNet perform very similarly when training different architectures, so they have cross-architecture versatility. Furthermore, our LoDM method achieves the best performance in almost all architectures, benefiting from its ability to hold more images in low-rank space at the same memory cost.

**Data Distribution Analysis**. We verify that the condensed data learned in the low-rank manifold can also capture the characteristics of the original data distribution. We first train a ConvNet model using all the original data as a feature extractor. Then, we input the original images, DC, DM, LoDC, and LoDM learned data (50 images per class) into the network to extract features and perform dimensionality reduction through T-SNE [54] to visualize. As shown in Fig. 1, we observe that both DC and LoDC cannot capture the distribution of raw data well, since they aim to perform dataset distillation with the goal of gradient matching. DM considers distribution matching as the distillation goal, which captures the data distribution effectively. LoDM inherits the properties of DM, particularly when the rank increases, e.g., in Fig. 1(d) and Fig. 1(e), the rank is 2 and 8, respectively.

**Hyper-parameter Analysis**. We analyze the effect of two hyperparameters, rank size $r \in \{1, 2, 4, 8\}$ and learning rate $\eta_{dc} \in \{0.1, 0.5, 1, 2, 5\}$, on CIFAR10 based on the LoDM method. Fig. 2(b) shows that when using the same number of images as DM, LoDM(SI) gradually achieves similar accuracy to DM in the high-dimensional pixel space, as the rank size increases. When the rank is 8, the performance is nearly identical to that of DM. We found that LoDM(SM) consistently outperforms DM when using the same memory, and the rank equal to 2 is a good choice based on empirical evidence. We also provide image visualizations generated by different ranks in the appendix. Fig. 2(c) shows that updating the generated data with a relatively large learning rate, such as 1 or 5, yields better results.

**Visualization of Images**. We visualize in Fig. 3 the condensed images generated by our LoDM method (rank $r = 2$) for the CIFAR10 and MNIST datasets, ten images per class. We observe that compared to the images condensed by DM in high-dimensional pixel space, the images we recover under the low-dimensional manifold will be sharper, but still capture the main features of the class, especially on the MNIST dataset. This further verifies that images are locally connected and have lower intrinsic dimensions, so dataset condensation in traditional pixel space may be inefficient.

## 4.2 Data Condensation for Continual Learning

In this section, we apply the low-rank dataset distillation plugin to continual learning tasks. We perform class-incremental learning with strict memory constraints, specifically using 10 images per class for the CIFAR10 dataset and 20 images per class for the CIFAR100 dataset. Based on Sec. 3.4,

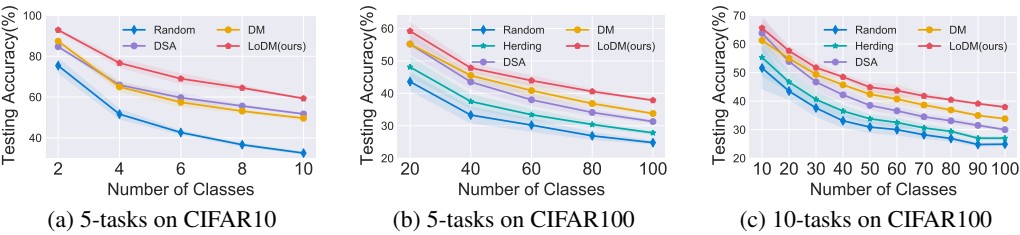

| (a) 5-tasks on CIFAR10 | (b) 5-tasks on CIFAR100 | (c) 10-tasks on CIFAR100 |

Figure 4: Test accuracy on the class-incremental learning task.

we combine the coreset method (Random, Herding) and various dataset distillation methods (DSA, DM, and our LoDM) into a simple and effective CL method GDumb [41]. We conduct experiments on two benchmark datasets, CIFAR10 and CIFAR100, where the CIFAR10 dataset is divided into 5 tasks, and the CIFAR100 dataset is divided into 5 tasks and 10 tasks, respectively. Based on Fig. 4, we observe that in the three subfigures (a-c), GDumb+LoDM achieves the best results. For example, on CIFAR100 with 5 tasks, the final accuracies of GDumb+Random, GDumb+Herding, GDumb+DSA, GDumb+DM, and GDumb+LoDM are 27.9%, 27.0%, 30.0%, 33.81%, and 37.92%, respectively. This suggests that our condensed data in a low-rank manifold is also meaningful for continual learning with limited memory.

# 5 Conclusion and Future Works

In this work, inspired by natural images that are locally connected and have low intrinsic dimensions, we propose a simple yet effective plugin that condenses a large dataset into a smaller dataset in a low-dimensional manifold. We apply this plugin to the existing dataset condensation methods and observe significant performance improvements while maintaining the same memory cost. Additionally, the analysis revealed that the dataset condensed in the low-dimensional manifold exhibits similar characteristics to the traditional high-dimensional pixel space dataset condensing method, such as matching the distribution of the large dataset and cross-architecture transferability. Furthermore, the plugin effectively addresses the issue of catastrophic forgetting in continual learning tasks. Our work has two directions for further improvement in the future: (i) Rank is a manually tuned hyperparameter in this paper. Therefore, how to adaptively assign the best rank to each dataset in the future is a feasible direction. (ii) We plan to apply the proposed dataset distillation plugin to more downstream tasks in the future, such as network architecture search, federated learning, and meta-learning.

## Acknowledgements

Li Shen was supported by the STI 2030—Major Projects (No. 2021ZD0201405). Enneng Yang and Guibing Guo were supported by the National Natural Science Foundation of China under Grants No. 62032013 and No. 61972078, the Science and technology projects in Liaoning Province (No. 2023JH3/10200005), and the Fundamental Research Funds for the Central Universities under Grants No. N2217004 and No. N2317002. Tongliang Liu was partially supported by the following Australian Research Council projects: FT220100318, DP220102121, LP220100527, LP220200949, and IC190100031.

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

# A  Implementation Details

**Experiment Details**. We train a model using condensed synthetic images and measure the Top-1 accuracy on raw testing images. For a fair comparison, our main experimental settings follow the settings of DC, DSA and DM. Experiments are conducted on a 3-layer convolutional network [46] (ConvNet-3) with 128 filters, without any specific emphasis. The batch size of the raw images is set to 256 during the matching process. For the DC [66], DSA [64], LoDC, and LoDM methods, the outer loop is set to 1 and the inner loop is set to 1 when using 1 image per class in all experimental datasets. When using 10 images per class, the outer loop is set to 10 and the inner loop is set to 50. In the MNIST and CIFAR10 datasets, when using 50 images per class, the outer loop is set to 50 and the inner loop is set to 10. For the DM [65] and LoDM methods, the loop is set to 20,000 in all experiments. The optimizers all use SGD. When optimizing the condensed data, the learning rate for DC/DSA/LoDC/LoDSA is set to 0.1 by default, and the learning rate for DM/LoDM is set to 1.0 by default. When using condensed data to train the network, the update times of the network are set to 1,000, and the learning rate is set to 0.01.

**Relationship between Memory and Rank**. Suppose the shape of each channel of a real image is $(H, W)$, where $H$ and $W$ are the height and width of the image, respectively. The memory consumed by traditional dataset distillation can be recorded as $N \times D \times H \times W$, where $N$ and $D$ represent the number of images and the number of channels. The memory consumed by our low-rank dataset distillation method is $N \times D \times (H \times r + r \times W)$, where $r$ is the rank size. For example, in the CIFAR10 dataset, both $H$ and $W$ are 32, and our $r$ is generally set to 2, which means that it can save 8 times the storage compared to traditional dataset distillation methods (i.e., DC, DSA and DM).

# B  Additional Experimental Results

## B.1  Experimental Analysis

**Compare with Other Adavanced Methods**. In our main paper, the low-rank plugin is mainly used in the traditional DC [66]/DSA [64] based on gradient matching and DM [65] based on distribution matching. Recently, a research work (i.e., MTT [8]) shows that better dataset condensation performance can be obtained based on finer-grained gradient trajectory matching. This is strictly orthogonal to our work. As shown in the Tab. 4, when our low-rank plugin is further applied to the MTT, our work achieved 58.7% accuracy on CIFAR10 (Img/Cls=1) and 31.0% on CIFAR100 (Img/Cls=1). This is a significant improvement compared to MTT, which is 46.3% and 24.3%.

In addition, related to our work are recent works on efficient dataset distillation [23, 67, 33, 13]. Among them, (i) IDC-I/IDC [23] and IDM [67] consider the local similarity of image pixels, and they partition and expand the synthetic image to obtain more images. This is orthogonal to our work, i.e., we can perform partitioning and expansion operations on images reconstructed by our low-rank plugin to further increase the number of images under the same memory constraints. For example, on the CIFAR10 dataset, compared with IDC-I/IDC, our low-rank versions (i.e., LoIDC-I/LoIDC) bring performance improvements of 12.5% and 6.6%, respectively. (ii) HaBa [33] inputs the bases into the Hallucinator networks to reconstruct condensed images, while RememberThePast [13] shares bases among all classes and constructs condensed images by addressing matrices. We note that their basis is also equal to the original image size. Therefore, under the same memory cost, we can use our low-rank plugin on their bases to obtain more images. For example, on the CIFAR10 dataset, our LoHaBa and LoRememberThePast improved by 17.8% and 4.0%, respectively, compared to Haba and RememberThePast in Tab. 4.

Table 4: Compare with other advanced dataset condensation methtods.

| | MTT | IDC-I | IDC | HaBa | RememberThePast |
|---|---|---|---|---|---|
| | 46.3% | 36.7% | 50.6% | 48.3% | 66.4% |
| CIFAR10 (Img/Cls=1) | LoMTT | LoIDC-I | LoIDC | LoHaBa | LoRememberThePast |
| | 58.7% | 49.2% | 57.2% | 66.1% | 68.4% |
| | MTT | IDC-I | IDC | HaBa | RememberThePast |
| | 24.3% | 16.6% | 24.9% | 33.4% | - |
| CIFAR100 (Img/Cls=1) | LoMTT | LoIDC-I | LoIDC | LoHaBa | LoRememberThePast |
| | 31.0% | 26.9% | 33.1% | 36.1% | - |

**Principal Component Analysis of Images**. In this section, we verify that both original images and images condensed by traditional dataset distillation methods (e.g., DSA [64] and DM [65]) have low-rank properties. Therefore, it is inefficient to perform dataset condensation directly in high-dimensional pixel space. Specifically, we randomly sample some images from the CIFAR10 dataset and further perform singular value decomposition (SVD) on each channel of these images. As shown in Fig. 5, the diagonal matrix represents the distribution of the average eigenvalues on all channels of all selected images. We can observe that both the original images (Fig. 5(a)) and the images synthesized by the traditional data distillation methods (e.g., DSA [64] in Fig. 5(b) and DM [65] in Fig. 5(c)) can be approximated by low rank, that is, the first few eigenvalues on the diagonal dominate. Therefore, it is reasonable to perform low-rank dataset distillation in this paper. Fig. 5(d) shows the distribution of eigenvalues of images produced by the proposed LoDM.

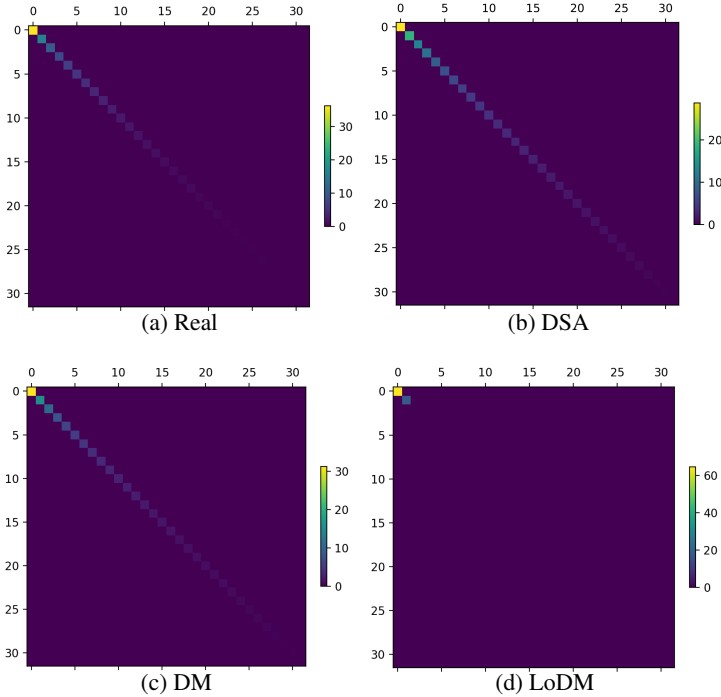

Figure 5: Principal component analysis of images on CIFAR10 dataset: (a) Real images; (b) Synthetic images of DSA; (c) Synthetic images of DM; and (d) Synthetic images of LoDM($r = 2$).

## B.2 Image Visualization

**Training Process of Synthetic Data**. Fig. 6 and Fig. 7 illustrate the intermediate state images of different ranks ($r = \{2, 8\}$) of our LoDM during the iterative process of synthesizing images on the CIFAR10 dataset. In addition, Fig. 10 and Fig. 11, we visualize the intermediate state images of different ranks ($r = \{2, 4\}$) of our LoDM in the iterative process of synthesizing images on the MNIST dataset. **Visualization of Real and Synthetic Images**. We provide more visualizations of the synthetic images from the different datasets: MNIST in Fig. 8, and CIFAR10 in Fig. 9. As the rank size increases, the synthesized image captures increasingly detailed features.

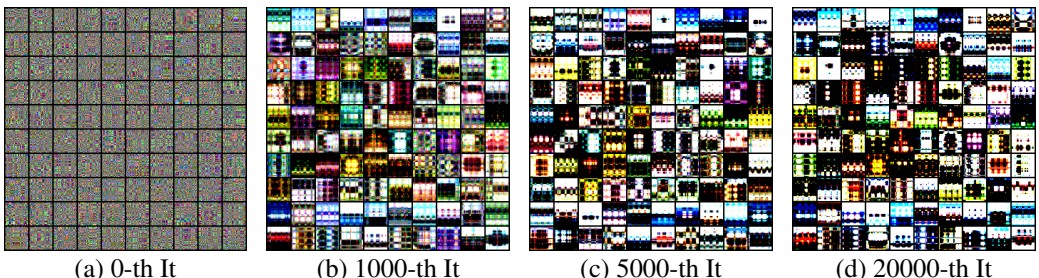

| (a) 0-th It | (b) 1000-th It | (c) 5000-th It | (d) 20000-th It |

Figure 6: Visualization of the synthetic image process of LoDM(r=2) on the CIFAR10 dataset.

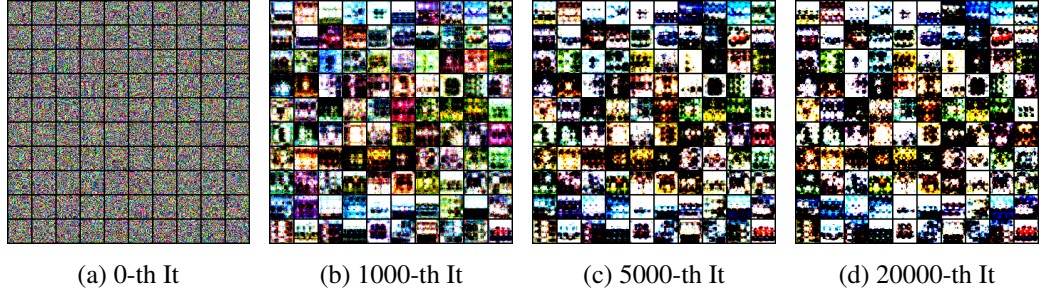

| (a) 0-th It | (b) 1000-th It | (c) 5000-th It | (d) 20000-th It |

Figure 7: Visualization of the synthetic image process of LoDM(r=8) on the CIFAR10 dataset.

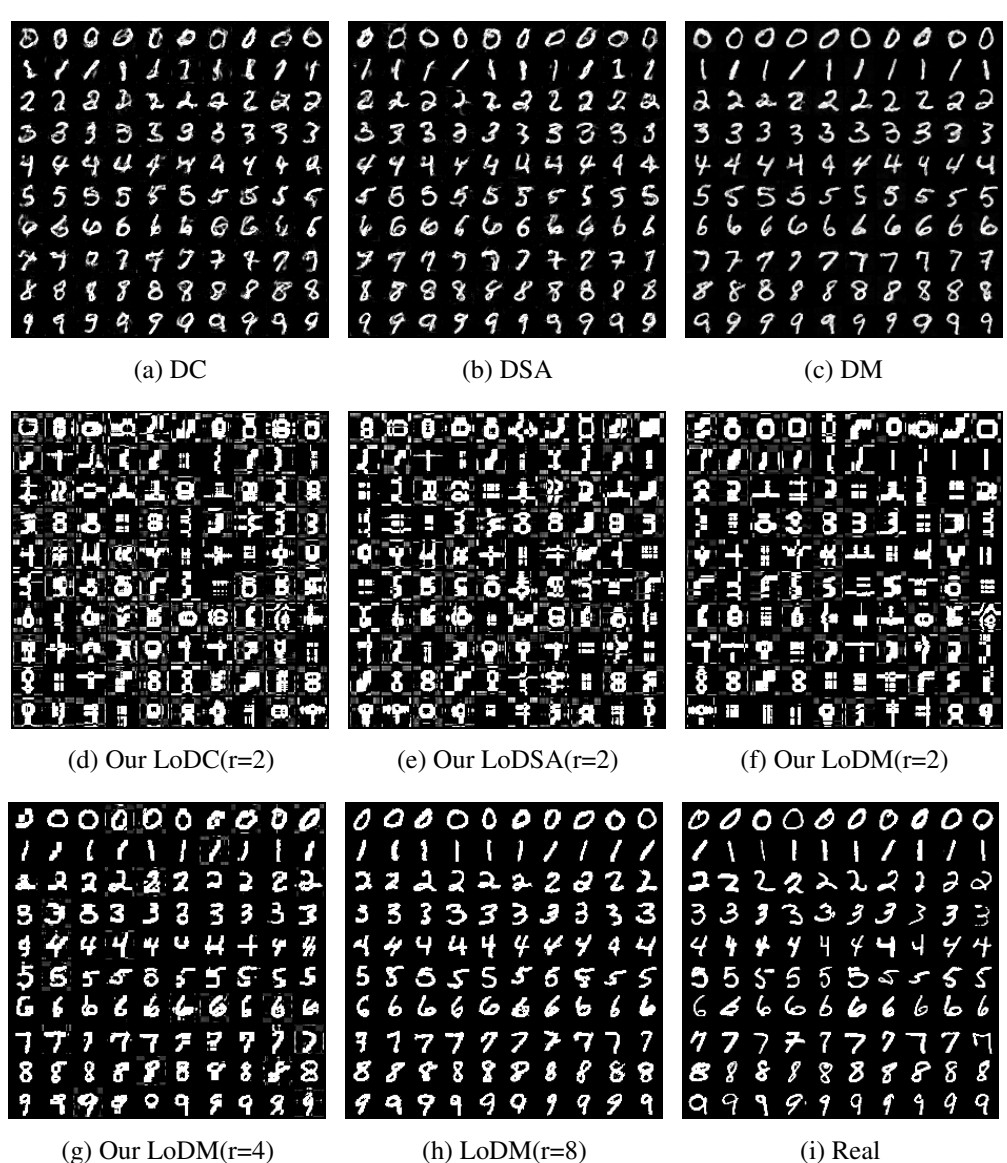

| (a) DC | (b) DSA | (c) DM |

| (d) Our LoDC(r=2) | (e) Our LoDSA(r=2) | (f) Our LoDM(r=2) |

| (g) Our LoDM(r=4) | (h) LoDM(r=8) | (i) Real |

Figure 8: Visualization on the MNIST dataset (10 Img/Cls).

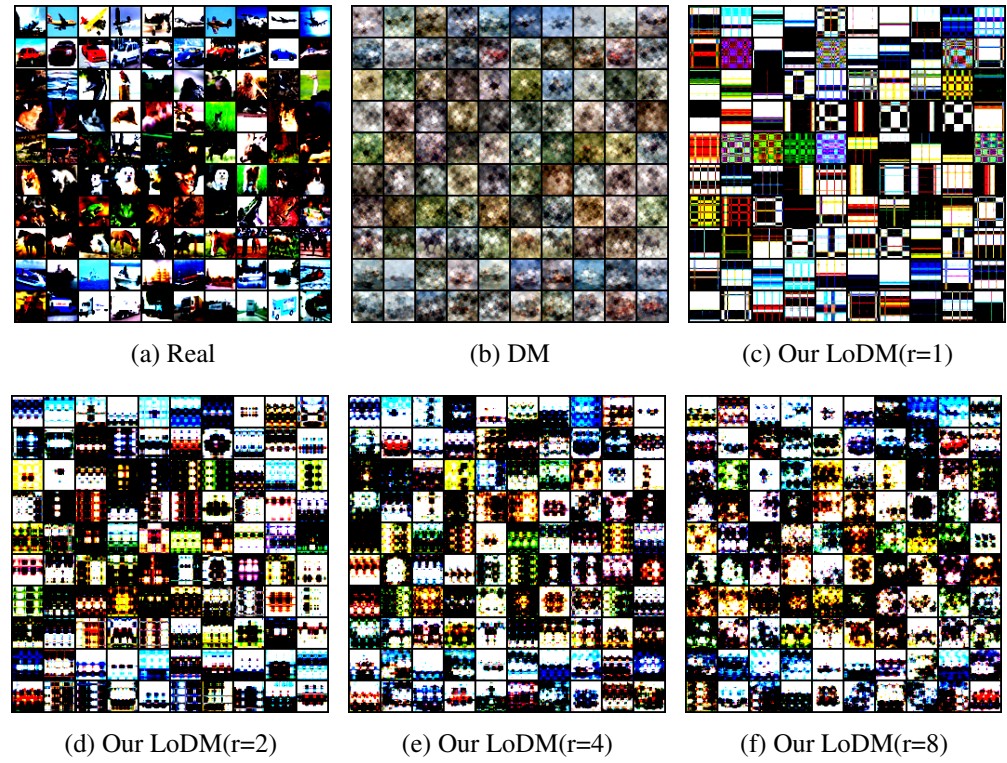

(a) Real

(b) DM

(c) Our LoDM(r=1)

(d) Our LoDM(r=2)

(e) Our LoDM(r=4)

(f) Our LoDM(r=8)

Figure 9: Visualization on the CIFAR10 dataset (10 Img/Cls).

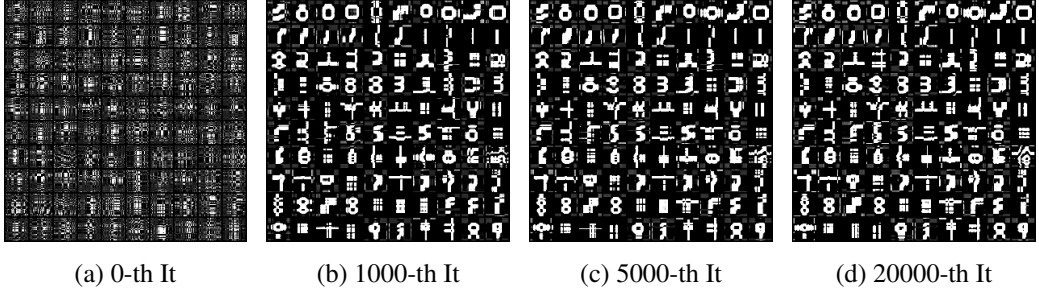

(a) 0-th It

(b) 1000-th It

(c) 5000-th It

(d) 20000-th It

Figure 10: Visualization of the synthetic image process of LoDM(r=2) on the MNIST dataset.

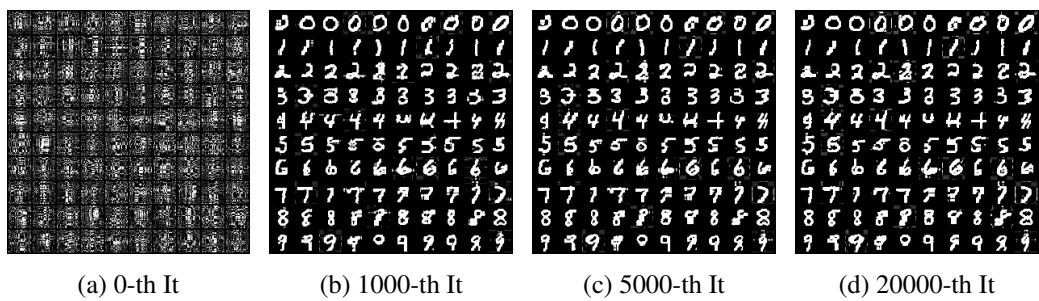

(a) 0-th It

(b) 1000-th It

(c) 5000-th It

(d) 20000-th It

Figure 11: Visualization of the synthetic image process of LoDM(r=4) on the MNIST dataset.

