# OpenReview forum: "An Efficient Dataset Condensation Plugin and Its Application to Continual Learning"
_NeurIPS.cc/2023/Conference — NeurIPS 2023 poster_

### Official Review · Reviewer_Lbw7 · 2023-07-05

**Soundness:** 3 good
**Presentation:** 3 good
**Contribution:** 3 good
**Rating:** 6
**Confidence:** 4

**Summary:**

The present paper studies the task of dataset condensation (DC) with a specific focus on its computational and condensation efficiency. The main idea is to condense the dataset in a low-rank representation space instead of in the high-dimensional pixel space. To do so, the author decomposes an image representation into two low-rank matrices, where the proposed method is orthogonal to the existing DC methods and can be readily integrated with existing DC methods. The author considers the gradient matching-based and distribution matching-based DC methods as examples to show how the proposed method can significantly improve them. Extensive experiments are conducted on common classification benchmarks, and the proposed method outperforms the state-of-the-art DC methods with large margins.

**Strengths:**

1. The idea of reducing the input distribution's dimensionality to increase the effective memory size and facilitate the DC is novel. Though leveraging the data's low-rank manifolds/inductive bias is not novel in general, as stated in lines 133-144, it is still interesting to see this idea can effectively work on DC, which is a surging interest in the community.

2. The paper is well-written, with clear motivation and excellent clarity. The technical details are easy to follow.

3. The proposed method can significantly improve the existing SOTA DC methods by a large margin, and it can also sufficiently benefit continual learning, which strongly demonstrates the significance and generality of the proposed method.

**Weaknesses:**

1. The main weakness is on the experimental protocol. Although the reviewer understands that the author wants to align with the existing methods in DC and choose the same input size for experiment, the input size of the MNIST (28 x 28), CIFAR100 (32 x 32), and TinyImageNet (64 x 64) are too small to be realistic. Given that the proposed method aims to reduce the input space dimensions, it is more favorable to see how effective the proposed method will be when the input size is large enough, e.g., at least 512 x 512. Only by doing so can we clearly see the benefits of the proposed method.

On the other hand, using realistic input size also provides us with a comprehensive understanding of the proposed method, especially its potential limitations. The reason is that we may intuitively expect that reducing a realistic size of the image to a low dimension (e.g., with rank size 2) can not retain the sufficient information and fine-grained visual context of the image. This is a clear limitation of the proposed method, and the author should definitively provide more analysis of it to inspire future study.

2. The author should at least provide the results of DC and DSA in Table 1 for the TinyImageNet, given that they may achieve SOTA in datasets like MNIST and CIFAR10. It would be helpful to see whether the proposed method significantly outperforms all the comparison methods in a more realistic dataset like TinyImageNet.

3. The result of the proposed method on MNIST is weird when Img/Cls is 10, as it is even worse than the original DM method. Can the author provide any plausible reason to explain it?

4. For the same memory setting in Table 2, how many images have been used by the proposed method? What if the comparison methods in Table 2 also leverage the same number of images as the one used in the same memory setup of the proposed method? Will the proposed method still outperform those compared methods?

5.  In Figure 1, we can observe that only when rank $r=8$ can the proposed method approximate the DM method's latent space structure. Then, how much memory has the proposed method reduced when rank $r=8$? Is this memory reduction still significant?

6.  In lines 284-286, the author states that the proposed method (rank $r=2$) reduces storage consumption by 7x, 8x, 8x, and 8x under different datasets. Please provide the exact number of storage between the proposed method and the comparison method.

7. In lines 322-326, the author states that "We observe that compared to the images condensed by DM in high-dimensional pixel space, the images we recover under the low-dimensional manifold will be sharper, but still capture the main features of the class, especially on the MNIST dataset." However, from Table 1, it is evident that the proposed method is much inferior to the compared methods on the MNIST dataset, which contradicts the claim above-mentioned. How to interpret this phenomenon?

Minor:

1. For subfigure (a) of Figure 1, how does the author obtain this T-SNE plot? Does the author directly plot the T-SNE over the pixel space or the feature embedding of the input images?

--------------------------------------------------------------------------------------------------------------------------------------------------------------

Final Acknowledgement on Sep 2nd:

Thank you so much for your detailed response. After reading all the reviewer's comments and the corresponding author's response, my concerns have been appropriately addressed.

**Questions:**

Please refer to the Weakness section for more details.

---

> ### Author Rebuttal · Authors · 2023-08-10
>
>
>  - **Advantages in high-resolution images:** Thanks for your suggestion, in fact our method works better when the original image resolution is higher. As shown in $\underline{\text{Fig.2 in the response}}$, we sampled a subset of ImageNet and tested the performance of DM and our LoDM at different resolutions $(64 \times 64, 128 \times 128, 256 \times 256, 512 \times 512)$. We can observe that the performance of DM does not change significantly at different resolutions, while LoDM, which considers low-rank characteristics, can save more information than DM at the same memory cost when the resolution is increased, so the effect continues to improve. In addition, the size of the rank is closely related to the number of images, although a low rank may not be able to restore the fine-grained information of a single high-resolution image, but it can save more images. As shown in $\underline{\text{Fig.2 in the original manuscript}}$, we empirically find that the performance across ranks is generally good.
>  - **TinyImageNet results:** We supplement the results of DC and DSA under TinyImageNet. For example, as shown in below table, when Img/Cls=1, the performance of DC and LoDC are $4.61\%$ and $6.8\%$, respectively; the accuracy of DSA and LoDSA are $4.79\%$ and $9.9\%$, respectively. Our method significantly improves the DC and DSA algorithms. We will update the results in the revised version.
>
> | DC | DSA | LoDC(Ours) | LoDSA(Ours)|
> | :------:| :------: | :------: | :------: |
> | 4.61%   | 6.8% | 4.79% | 9.9% |
>
>  - **Reason for poor performance on MNIST(Img/Cls=10):** Thanks for pointing this out. The reason we do not perform well on MNIST(Img/Cls=10) is that we strictly follow the hyperparameters on DM, which are fine-tuned for DM. When we simply adjust the hyperparameters (learning rate or rank) of our LoDM in MNIST, LoDM has achieved performance improvement. For example, the accuracy of LoDM in MNIST (Img/Cls=10) is $97.7\%$, which exceeds $96.5\%$ of DM. Similarly, CIFAR10(Img/Cls=100) has an accuracy of $64.69\%$ after fine-tuning, which exceeds $63.0\%$ of DM; for CIFAR100(Img/Cls=10), the accuracy of LoDSA is $33.7\%$, while that of DSA is $32.3\%$.
>
> | Dataset | Base method | Our method(Update version) |
> | :------:| :------: | :------: |
> | MNIST(Img/Cls=10)   | DM: 96.5% | LoDM: 97.7% |
> | CIFAR10(Img/Cls=100)   | DM: 63.0% | LoDM: 64.69% |
> | CIFAR100(Img/Cls=10)   | DSA: 32.3% | LoDSA: 33.7% |
>
>  - **Number of images in Table 2:** Suppose the shape of each channel of the image is $(H, W)$, where $H$ and $W$ are the height and width of the image, respectively. The memory consumed by traditional dataset distillation (DC/DSA/DM) can be recorded as $N \times D \times H \times W$, where $N$ and $D$ represent the number of images and the number of channels. The memory consumed by our low-rank dataset distillation method is $N \times D \times (H \times r+r \times W)$, where $r$ is the size of the rank. Therefore, under the same memory, when the traditional method stores an image in MNIST($28 \times 28$), CIFAR10($32 \times 32$), CIFAR100($32 \times 32$), if $r=2$, then our LoDC(SM)/LoDSA( SM)/LoDM(SM) can store $7, 8, 8$ pictures respectively. On the contrary, when using the same image, LoDC(SI)/LoDSA(SI)/LoDM(SI) is slightly worse than DC/DSA/DM, but they also reduce the memory consumption by $7, 8, 8$ respectively.
>  - **Memory consumption:** When $rank=8$ in  $\underline{\text{Fig.1 in the original manuscript}}$, on the MNIST dataset, our method saves $(28/16=1.75)$ times the memory for the same number of images.
>  - **Relationship between memory and rank:** Please refer to the reply to your 4-th question.
>  - **MNIST performance issues:** Please refer to the reply to your 3-th question, our method is significantly better than the traditional method on the MNIST dataset after simple fine-tuning the hyper-parameters.
>  - **T-SNE visualization:** We first trained a CNN network using the complete MNIST raw data set, then used it as a feature extractor to extract high-dimensional features of the original image or distilled image, and finally reduced the high-dimensional features through T-SNE visualization to obtain the results of $\underline{\text{Fig.1 in the original manuscript}}$.

---

> > ### Comment · Reviewer_Lbw7 · 2023-08-21
> >
> > Thank you so much for your detailed response. After reading all the reviewer's comments and the corresponding author's response, I am upgrading my score to 6 as my concerns have been appropriately addressed.

---

### Official Review · Reviewer_VZYQ · 2023-07-06

**Soundness:** 4 excellent
**Presentation:** 4 excellent
**Contribution:** 4 excellent
**Rating:** 7
**Confidence:** 4

**Summary:**

The authors propose an efficient dataset condensation (DC) plugin that utilizes a low-dimensional manifold, namely two low-rank matrices, to match the large raw and smaller synthetic condensed dataset. This aims to reduce inefficient condensation when matching in image pixel space. They show that their method is modular, and easy to incorporate into existing DC methods, and is effective, improving model performance when trained on the smaller synthetic dataset. Their method reduces catastrophic forgetting when the entire raw dataset is not available for replay, and reduces privacy leakage from the initial raw dataset.

After reading the authors rebuttal and discussing with them, as well as seeing the discussion with other reviewers, I will decrease my score to 7.

**Strengths:**

This work makes nice use of the structure implicit in natural images, namely local connectivity and similarity of nearby pixels and the low intrinsic dimensionality of natural images, to improve dataset condensation. They propose a simple but effective plugin to any modern DC method that performs DC in a low-dimensional manifold as opposed to in pixel space, significantly improving learning efficiency and memory consumption while maintaining downstream performance of models trained on the condensed datasets. The authors also demonstrate the benefit of their low-rank approach for continual learning, an intuitive application as memory is often constrained in continual learning frameworks. The paper is clearly written, the method well-described, and the analysis is thorough. I particularly like the modularity of the method, and that it is applicable and useful across DC methods.

**Weaknesses:**

It’s interesting that the synthetic digits in Figure 3d no longer always resemble the real categories. What are potential risks associated with this? Is it possible that selecting a condensation dataset via model selection on a fixed set of data could have downstream implications if this model is used on real-world data or data under a slightly different distribution? How would this method fare when datasets are imbalanced or biased, as is often the case in real-world settings? The datasets considered all are uniform and IID, unless I’m mistaken?

Nits:

The images in figure 1 and figure 2 should be labeled with (a) (b) etc for easier legibility

**Questions:**

Why do the authors think their method underperforms on MNIST in table 1, which has perhaps the lowest-dimensionality of any of the considered datasets?

**Limitations:**

I would like to see a discussion of failure modes of limitations of the method, where there any cases that surprised the authors where the model fails? How might this perform in practice, where categories of interest are often highly imbalanced and data is often biased or, in the case of continual learning, where distributions shift over time?

---

> ### Author Rebuttal · Authors · 2023-08-10
>
>
> - **The impact of synthetic data:** Indeed, in $\underline{\text{Fig.3(d) in the original manuscript}}$, it seems intuitively that the generated image is no longer similar to the real image, but from the perspective of model learning, the impact of this difference may be weak. On the one hand, as shown in the $\underline{\text{Fig.1 (a),(d) and (f) in the original manuscript}}$, traditional dataset distillation and the proposed low-rank dataset distillation can well preserve the distribution close to the real images. On the other hand, when there are downstream tasks, such as in the continual learning task shown in $\underline{\text{Fig.4 in the original manuscript}}$, the data distribution of the new task/class is significantly different from the old task/class. In this case, the method of dataset distillation still performs significantly better than the model trained on the real data (i.e., Random and Herding).
> - **Dataset distillation under imbalanced dataset:** Yes, the original experimental datasets are class balanced. In order to verify the effect when the category data distribution is unbalanced, we sample the data of the original 10 classes of CIFAR10, and the 10 classes respectively sample the original data of $1.0, 0.9, 0.8, ..., 0.1$, thus constructing an imbalanced-CIFAR10. We perform dataset distillation on this imbalanced dataset, and as shown in $\underline{\text{Fig.3(a) in the response}}$, the accuracy of dataset distillation methods does not decrease significantly on the imbalanced dataset. This is because the data set is performed class by class (refer to the fifth line of $\underline{\text{Alg.1 and Alg.2 in the original manuscript}}$ (that is, *for class $c = 1, 2, ..., |Y|$ do*, where $|Y|$ represents the total number of classes), and the data imbalance between classes will not affect the accuracy of distillation.
> - **Subfigure identification:** We will add subfigure identification as you suggested.
> - **Reason for poor performance on MNIST:** As shown in $\underline{\text{Tab.2 in the original manuscript}}$, our LoDC and LoDSA have significant performance improvements compared to DC and DSA. The reason why LoDM is not as effective as DM when Img/Cls=10 in $\underline{\text{Tab.1 in the original manuscript}}$ is that we strictly follow the hyperparameter settings of DM, which are obtained by searching on DM. As shown in bellow table, when we fine-tune the hyperparameters of LoDM (learning rate or rank), LoDM achieves $97.7\%$ performance, which is better than $96.5\%$ of DM. Similarly, after simply performing hyper-parameters search, our LoDM method also achieves $64.6\%$ accuracy on CIFAR10(Img/Cls=50), which is better than DM's $63.0\%$.
>
>
> | Dataset | Base method | Our method(Update version) |
> | :------:| :------: | :------: |
> | MNIST(Img/Cls=10)   | DM: 96.5% | LoDM: 97.7% |
> | CIFAR10(Img/Cls=100)   | DM: 63.0% | LoDM: 64.69% |
> | CIFAR100(Img/Cls=10)   | DSA: 32.3% | LoDSA: 33.7% |
>
> - **The reason why the method fails:** As mentioned above, our low-rank version (such as LoDM) strictly follows the hyperparameter setting of the base method LoDM), and we found that after simply adjusting the hyperparameters of the low-rank method, the proposed method is better than the traditional method on all datasets.
> - **Continual Learning with imbalanced data:** Based on the imbalanced dataset defined in the 2-th question for your reply, we performed a continual learning task on this dataset, which is the more challenging "data imbalance and distribution drift". The experimental results are shown in $\underline{\text{Fig.3(b) in the response}}$, and there is no significant difference in accuracy when we perform continual learning under the imbalanced version of CIFAR10 and the balanced version of CIFAR10. This is because the dataset distillation is performed class by class (refer to the fifth line of $\underline{\text{Alg.1 and Alg.2 in the original manuscript}}$), so the data imbalance between classes will not affect the accuracy of the dataset distillation. Further, the data of each class is distilled into a fixed number(Img/Cls=20) of synthetic images, so it does not affect the performance under continual learning.

---

> > ### Comment · Reviewer_VZYQ · 2023-08-19
> > **Slightly reducing my score after reading the other reviews**
> >
> > Thank you to the authors for the clarifications provided in their rebuttal, and for the additional discussion and comparison to data parameterization methods in response to reviewer FpMf. After reading the other reviews and some of the related work, I agree that the novelty of this method in comparison to prior work is less significant that I had initially interpreted, so I will decrease my score to a 7. However, I think this paper still clears the bar for acceptance, particularly given the authors additional contextualization of how it is similar and different to other prior work qualitatively and quantitatively.

---

### Official Review · Reviewer_FpMf · 2023-07-07

**Soundness:** 2 fair
**Presentation:** 3 good
**Contribution:** 1 poor
**Rating:** 4
**Confidence:** 4

**Summary:**

This paper introduces a novel dataset distillation technique, referred to as LoDM.  Instead of storing data in pixel space, LoDM propose to store each condensed image in a low dimensional space which is represented by multiplication of two matrixes. By decreasing the memory budget allocated for each image, LoDM significantly reduces memory cost. LoDM can be easily combined with SOTA data condensation loss. The evaluation results on four datasets show that LoDM outperforms the presented baselines.

**Strengths:**

1.	The paper proposes a novel data parameterization method to store the condensed information, which improves the storage efficiency for data condensation.
2.	The evaluation results show that LoDM outperforms other baselines which store condensed information in images.
3.	The writing is clear and easy to follow.

**Weaknesses:**

Many thanks to paper, I really enjoy reading the paper and the idea of LoDM, but I observed a notable absence in the discussion on a highly relevant area - data parameterization, within the realm of data condensation…

In fact, LoDM could be viewed as a variant of the data parameterization method. Storing condensed information in other formats rather than single image is not new for data condensation, which is known as data parametrization. Several data parameterization methods[1][2][3] have been proposed, yet this paper fails to discuss or compare them. Moreover, these data parameterization methods deliver better performance than LoDM.

Without a thorough discussion and comparison with these related works, it becomes challenging to truly evaluate the novelty and effectiveness of LoDM.

[1] Kim, Jang-Hyun, et al. "Dataset condensation via efficient synthetic-data parameterization." International Conference on Machine Learning. PMLR, 2022.

[2] Liu, Songhua, et al. "Dataset distillation via factorization." Advances in Neural Information Processing Systems 35 (2022): 1100-1113.

[3] Deng, Zhiwei, and Olga Russakovsky. "Remember the past: Distilling datasets into addressable memories for neural networks." Advances in Neural Information Processing Systems 35 (2022): 34391-34404.

**Questions:**

Compared to SOTA data parameterization methods, what is the advantage of LoDM?

**Limitations:**

Yes.

---

> ### Author Rebuttal · Authors · 2023-08-10
>
>
> - **Relationship to works [1,2,3]:** Thanks for your suggestions for related works. Although our work and [1,2,3] seem to be data parameterization methods, the motivation of our work is fundamentally different from theirs: IDC[1] constructs multi-formation synthetic images by upsampling, using neighborhood similarity of image pixels. Factorization[2] inputs the base into multiple hallucination networks to generate multiple images, and uses contrastive learning to enhance the diversity of images to enhance performance. RememberThePast[3] shares a set of bases among images of all classes, exploiting image-to-image correlations. In summary, these works [1-3] all consider the image characteristics at the pixel level, and do not consider the global low-rank characteristics of the image. As shown in $\underline{\text{Fig.1(a) and Fig.1(b) in the response}}$, we found that both real images and images generated by traditional dataset distillation method(e.g., DM) are of low rank, that is, a small number of eigenvalues dominate. Based on this, we propose to perform dataset distillation in low-rank manifolds, so our method is complementary to these works. We proposed low-rank plugin can be applied to any existing dataset distillation methods, including the above three methods. Specifically, when combined with IDC[1], we can separately learn a set of multi-formation images and a set of low-rank images to simultaneously consider the local similarity and low-rank properties of images. When combined with Factorization[2] and RememberThePast[3], we can decompose the parameterized basis into two low-rank vectors, thus storing more basis at the same memory cost. For example, as shown in bellow table, when our low rank is combined with Factorization[2], the accuracy on the CIFAR10 (Img/Cls=1) dataset is improved from 48.3% to 66.1%. We believe that applying our low-rank plugin to IDC[1] and RememberThePast[3] will further improve their performance as well. In our revision, we will discuss the mentioned papers[1,2,3] in detail.
>
> | Factorization[2] | LoFactorization(Ours)|
> | :------:| :------: |
> | 48.3%   | 66.1% |
>
> - **Performance comparison with works [1,2,3]:** In our paper, the low-rank plug-in is mainly used in the traditional DC/DSA based on gradient matching and DM based on distribution matching. However, recent works (e.g. Factorization[2]) are mainly based on better-performing gradient-trajectory-matching-based methods (MTT). As shown in bellow table, when our low-rank plug-in is further applied to the MTT, our work achieved 58.7% accuracy on CIFAR10 (Img/Cls=1). This is a significant improvement compared to MTT (46.3%), DIC[1] (50.6%), Factorization[2] (48.3%), second only to RemeberThePast[3] (66.4%). However, our low-rank plug-in can be further applied to [1,2,3], for example, when combined with Factorization[2], our method is further improved to 66.1%. Due to time constraints, the code of our low-rank method applied to RememberThePast[3] is still running, and we will present it after subsequent executions are complete. Intuitively, our low-rank plugin allows us to store more images/basis than RememberThePast[3] in the same memory, so we think our method(LoRememberThePast) will further improve compared to RememberThePast[3].
>
>
> | MTT | IDC[1] | Factorization[2] | RememberThePast[3] | LoMTT(Ours)| LoFactorization(Ours)|
> | :------:| :------: | :------: | :------: | :------: | :------: |
> | 46.3% | 50.6%   |  48.3%   | 66.4%   | 58.7% | 66.1% |
>
> - **Advantage of our method:** We found that the images generated by the dataset distillation method have a global characteristic of low rank, which has not been considered in previous image parameterization methods, and their core idea is to use the local characteristics of image pixels. Furthermore, our proposed method based on this property is simple-yet- effective, and can be combined with existing methods as a versatile plug-in to save more information of the original dataset at the same memory cost, greatly improve the performance of traditional dataset distillation methods.

---

> > ### Comment · Reviewer_FpMf · 2023-08-16
> > **Thanks for the response**
> >
> > Thank the authors for explaining the difference between LoDM and other data parameterization work, and I understand that LoDM is based on a different insight compared to other data parameterization methods. The authors claim that LoDM can be combined with other data parameterization methods. However, my major concern is that no extensive evaluation supports this claim.
> >
> > From my perspective, LoDM is a type of data parameterization method. *I believe that it is necessary to conduct an extensive comparison with SOTA data parameterization methods on multiple benchmarks and settings (it could be LoDM alone or combining LoDM with other methods).* Without these evaluations, it is not convincing enough to prove the effectiveness of LoDM. Based on this, I decided to keep my rating unchanged at this point.

---

> > > ### Author Response · Authors · 2023-08-18
> > >
> > > Thank you very much for recognizing the motivation and innovation of our work. To verify the effectiveness of our low-rank method under various dataset distillation methods, we conducted additional verification experiments on CIFAR10(Img/Cls=1). Specifically, we introduced comparisons with MTT [4], a method based on training trajectory matching, as well as methods employing data parameterization, namely IDC-I (Uniform formation) [1], IDC (Multi-scale formation) [1], and Factorization [2]. The experimental results are shown in the table below. After combining our low-rank method with the four baseline methods, all of them significantly improve their performance. This improvement is attributed to the increased number of images that can be stored within the same memory capacity after using our low-rank approach.
> > >
> > >
> > > Specifically, our implementation details are as follows:
> > > - When combined with MTT [4], we optimize two sets of low-rank parameters according to the training trajectory matching, instead of optimizing a set of parameters for the original image size as MTT does. Based on the following experiments, we can see that LoMTT(Ours) can significantly boost the performance of MTT methods, with up to 12.4% accuracy.
> > > - When combining IDC-I [1] and IDC [1], we execute an upsampling operation on the data composed of two sets of low-rank vectors to obtain a set of multi-formation synthetic images, instead of directly performing an upsampling operation on a set of raw image-sized data to obtain a set of multi-formation composite images. According to the results in the table below, compared with IDC-I and IDC, our approaches LoIDC-I(Ours) and LoIDC(Ours) yield accuracy improvements of 15.2% and 7.3%, respectively.
> > > - When combined with Factorization [2], we use two sets of low-rank vectors to represent the basis of the synthetic image instead of using the original image-sized basis like Factorization. Compared to Factorization, our approach LoFactorization(Ours) brings a 17.8% performance improvement.
> > >
> > >
> > > | Method | ACC |
> > > | :------:| :------: |
> > > | | |
> > > | MTT [4] | 46.3% | |
> > > | **LoMTT(Ours)** | **58.7%** |
> > > | | |
> > > | IDC-I [1] | 36.7% |
> > > | IDC-I(reproduction) | 34.0% |
> > > | **LoIDC-I(Ours)** | **49.2%** |
> > > | | |
> > > | IDC [1] | 50.6% |
> > > | IDC(reproduction) | 49.9% |
> > > | **LoIDC(Ours)** | **57.2%** |
> > >  | |
> > > | Factorization [2] | 48.3% |
> > > | **LoFactorization(Ours)** | **66.1%** |
> > >
> > > Since the implementation of the data parameterization method RememberThePast [3] requires more training iterations, we are currently debugging and training it, along with the version that incorporates our low-rank approach. As soon as new results become available, we will promptly provide updates.
> > >
> > >
> > > **In summary, based on the above explanations and experimental results, we can conclude that our proposed dataset condensation method is orthogonal with the mentioned four SOTA methods and can be served as a strong plugin for existing SOTA data condensation approaches to further boost their performance.**
> > >
> > >
> > > We hope these explanations and experiments can solve your concerns. All the discussions will be incorporated into our revision. If you have further questions, please let us know.
> > >
> > >
> > >
> > > References:
> > >
> > > [1] Kim, Jang-Hyun, et al. "Dataset condensation via efficient synthetic-data parameterization." International Conference on Machine Learning. PMLR, 2022.
> > >
> > > [2] Liu, Songhua, et al. "Dataset distillation via factorization." Advances in Neural Information Processing Systems 35 (2022): 1100-1113.
> > >
> > > [3] Deng, Zhiwei, and Olga Russakovsky. "Remember the past: Distilling datasets into addressable memories for neural networks." Advances in Neural Information Processing Systems 35 (2022): 34391-34404.
> > >
> > > [4] Cazenavette, George, et al. "Dataset distillation by matching training trajectories." IEEE/CVF Conference on Computer Vision and Pattern Recognition. 2022.

---

> > > > ### Author Response · Authors · 2023-08-20
> > > >
> > > > We further added a comparison with RememberThePast [3] on the CIFAR10 dataset, and the results shown in the table below show that our low-rank plugin helps to further substantially improve the performance of RememberThePast.
> > > >
> > > > In terms of implementation, RememberThePast shares a set of bases of the original image size among all classes, and combines these bases by addressing matrices to obtain the synthetic image. After introducing our low-rank plugin, we can combine two sets of low-rank representation bases by addressing matrices, which will allow us to store more bases at the same memory cost, thereby further improving the performance of dataset distillation. However, since the complete RememberThePast [3] (50,000 iterations is set in the paper) is time-consuming and complete results will take several days, we currently report the results of multiple intermediate iterations (iterations = \{200, 1000, 2000, 3000\}). As shown in the table below, at all iterations, the results of our LoRememberThePast (Ours) are better than RememberThePast. For example, when the iterations are 200, 1000, 2000, and 3000, our method improves by 6.2%, 3.8%, 3.8%, 3.9% respectively.
> > > >
> > > > | Iteration | 200 | 1000 | 2000 | 3000 |
> > > > | :------:| :------: |  :------: |  :------: |  :------: |
> > > > | | | | | |
> > > > | RememberThePast[3] | 48.1% | 57.6% | 58.5% | 59.9% |
> > > > | **LoRememberThePast(Ours)** | **54.3%** | **61.4%** | **62.3%** | **63.8%** |
> > > >
> > > > **The above experimental results further show that our low-rank plugin is also orthogonal to RememberThePast.**
> > > >
> > > > We hope these supplementary experiments address your concerns, and we will also add the full results in the revised version. If you still have questions, please discuss with us before the deadline.
> > > >
> > > >
> > > > Thanks again for your time.
> > > >
> > > > Best,
> > > >
> > > > Authors.

---

> > > > > ### Author Response · Authors · 2023-08-21
> > > > >
> > > > > Dear Reviewer FpMf,
> > > > >
> > > > > We further update the latest comparison results with RememberThePast [3]. As shown in the table below, at the 8000th and 12000th iterations, our LoRememberThePast(Ours) improved by 3.4% and 3.9% compared to RememberThePast, respectively.
> > > > >
> > > > > | Iteration |  8000 | 12000 |
> > > > > | :------:| :------: |  :------: |
> > > > > | | | |
> > > > > | RememberThePast[3] | 61.9% | 62.6% |
> > > > > | **LoRememberThePast(Ours)** | **65.3%** | **66.5%** |
> > > > >
> > > > > **In summary, all experimental results consistently show that our proposed low-rank distillation plugin is orthogonal to existing data parameterization work [1,2,3], and it can be combined with existing SOTA dataset distillation methods further improve performance.**
> > > > >
> > > > > We sincerely appreciate your help in improving our work during this process, and we kindly ask you to evaluate our work again.
> > > > >
> > > > >
> > > > > [1] Kim, Jang-Hyun, et al. "Dataset condensation via efficient synthetic-data parameterization." International Conference on Machine Learning. PMLR, 2022.
> > > > >
> > > > > [2] Liu, Songhua, et al. "Dataset distillation via factorization." Advances in Neural Information Processing Systems 35 (2022): 1100-1113.
> > > > >
> > > > > [3] Deng, Zhiwei, and Olga Russakovsky. "Remember the past: Distilling datasets into addressable memories for neural networks." Advances in Neural Information Processing Systems 35 (2022): 34391-34404.
> > > > >
> > > > >
> > > > > Thanks again for your time.
> > > > >
> > > > > Best,
> > > > >
> > > > > Authors.

---

> > > > > > ### Author Response · Authors · 2023-08-21
> > > > > >
> > > > > > Dear Reviewer FpMf,
> > > > > >
> > > > > > Thanks for your valuable comments and suggestions. We have provided a bunch of experiments in the rebuttal to show that our proposed method is orthogonal with existing data parameterization methods and they can be integrated to achieve new SOTA results for dataset condensation.
> > > > > >
> > > > > > Could you please check the experimental evaluation in the rebuttal?
> > > > > >
> > > > > > Best,
> > > > > >
> > > > > > Authors

---

> > > > > > > ### Comment · Reviewer_FpMf · 2023-08-22
> > > > > > > **Thanks again for further discussion on the work**
> > > > > > >
> > > > > > > The orthogonal explanation addressed some of my concerns about the lack of comparison and discussion on related data parameterization methods. And the evaluation results on CIFAR10 1IPC partially support this claim.
> > > > > > >
> > > > > > > I decided to increase my rating to borderline reject. The reason that I didn't further increase the rating is that I think that a more extensive evaluation on more datasets and memory budgets can better support this claim.
> > > > > > >
> > > > > > > Thank the authors again for the explanation and discussion.

---

> > > > > > > > ### Author Response · Authors · 2023-08-22
> > > > > > > >
> > > > > > > > Dear Reviewer FpMf,
> > > > > > > >
> > > > > > > > Thank you very much for your further support of our work, we will supplement the orthogonal experiments on other datasets as we promised. CIFAR10 is a very representative benchmark dataset in dataset distillation, while our method is a general plugin for dataset distillation. Just as our plugin consistently improved existing dataset distillation methods on multiple datasets in Table 1 in the manuscript, we believe that the performance improvement of our plugin on existing data parameterization methods on the CIFAR10 dataset should be expected to result in similar performance gains for other datasets.
> > > > > > > >
> > > > > > > > Thank you again for your suggestions and contributions to our work.
> > > > > > > >
> > > > > > > >
> > > > > > > > Best,
> > > > > > > >
> > > > > > > > Authors

---

> > ### Comment · Area_Chair_Mbwq · 2023-08-20
> > **Is the Evaluation Sufficient**
> >
> > Dear Reviewer FpMf,
> >
> > Regarding your comments about the comparison with data parameterization methods, the authors have added a bunch of experiments in the rebuttal.
> >
> > Could you please check the experimental evaluation at your earliest convenience?
> >
> > AC

---

### Official Review · Reviewer_Lynm · 2023-07-08

**Soundness:** 3 good
**Presentation:** 3 good
**Contribution:** 2 fair
**Rating:** 5
**Confidence:** 3

**Summary:**

This paper proposed a low-rank technique to reduce the storage of the compressed dataset. It successfully applied the LoRA technique in DC and DD problems, and also apply it to the CL problem.

After applying the LoRA technique, the method can still achieve similar performance compared to the original DM.

**Strengths:**

++ It is valuable and meaningful to reduce the storage of the synthetic data for dataset distillation.
++ The LoRA trick is successfully applied to various conditions.

**Weaknesses:**

-- The paper looks like directly applies the LoRA trick to the DC and DM methods.  Could the author give more explanation on how they adjust the LoRA for the data distillation task? What are the challenges they face and how do they make changes to overcome them?

-- The method names are not consistent, LoDM and LoraDM are used in the main paper.

-- It will be better if more analysis can be added to discuss how the proposed method makes use of the insight of "natural images that are locally connected and have low intrinsic dimensions".

--It will be better to discuss the relationship between the saved computations/ storage and the different ranks.

**Questions:**

Please see the weakness.

**Limitations:**

The author has discussed the limitation and social impact in the supplementary.

---

> ### Author Rebuttal · Authors · 2023-08-10
>
> - **The significance of our low-rank method:** As described in our related work, low-rank is a long-standing and fundamental research point in the field of machine learning. LoRA is not the first work to study low-rank. The contribution of LoRA is to consider the characteristics of low intrinsic dimension when fine-tuning the parameters of large language models, thus proposing a method for parameter-efficient tunings. Our work is the first to consider the low-rank characteristics of images in the field of dataset distillation. As shown in $\underline{\text{Fig.1(a) and Fig.1(b) in the response}}$, we found that both the real image (Fig.1(a)) and the image generated by the traditional dataset distillation method (Fig.1(b)) are of low rank , that is, a small number of eigenvalues dominate. Based on this strong motivation, we propose to perform dataset distillation directly in the low-rank manifold, which exploits low-rank properties to store more raw data information with as little space as possible. Our proposed low-rank dataset distillation plugin is general and can be combined with arbitrary dataset distillation methods to significantly improve their performance, for example in the table below, we compared the baseline method on CIFAR10 (Img/Cls=1). There are very significant improvements. We believe that filling this research gap in the dataset distillation is significant.
>
> | DC | LoDC(Ours)| DSA | LoDSA(Ours)| DM | LoDM(Ours)| MTT | LoMTT(Ours)|
> | :------:| :------: |:------:| :------: |:------:| :------: |:------:| :------: |
> | 28.3% |  35.2% | 28.8% |  41.0% | 26.0% |  43.8% | 46.3% |  58.7% |
>
> - **Consistency of method names:** Thanks, we fixed the issue.
> - **Insights of our approach:** In $\underline{\text{Fig.1(a) and Fig.1(b) in the response}}$, we performed principal component analysis on the real and synthetic images and found that a small number of eigenvalues on the diagonal are dominant. This result shows that natural images are low-rank, and images condensed by traditional dataset distillation method(i.e., DM) also low-rank. Based on this motivation, we propose a general low-rank dataset distillation plugin, which directly learns low-rank synthetic images instead of learning synthetic images in the original high-dimensional pixel space.
> - **Relationship between memory and rank:** Suppose the shape of each channel of a real image is $(H, W)$, where $H$ and $W$ are the height and width of the image, respectively. The memory consumed by traditional dataset distillation can be recorded as $N \times D \times H  \times W$, where $N$ and $D$ represent the number of images and the number of channels. The memory consumed by our low-rank dataset distillation method is $N  \times D \times (H  \times r + r  \times W)$, where $r$ is the size of the rank. For example, in the CIFAR10 dataset, both $H$ and $W$ are $32$, and our $r$ is generally set to $2$, which means that it can save $8$ times the storage compared to traditional dataset distillation methods. It is obvious that our method improves significantly when the original image resolution is higher. In $\underline{\text{Fig.2 in the response}}$, we give an example of images of different resolutions $(64 \times 64, 128 \times 128, 256 \times 256, 512 \times 512)$ on a subset of ImageNet, which effectively verifies this point. We will provide a detailed analysis of the relationship between memory and rank in the revised version.

---

> > ### Comment · Reviewer_Lynm · 2023-08-20
> > **Replying to the Authors**
> >
> > Figure 1 and the explanations fill the gap between the method and the motivation.
> > I agree to improve my rating after reviewing all the review comments and the authors' replies. The author should modify the motivation and include discussions in the next version.

---

> > > ### Author Response · Authors · 2023-08-21
> > >
> > > Dear Reviewer Lynn,
> > >
> > > Thank you for your suggestions on our work, we will revise the discussion in the next version. We kindly remind you that the score has not been updated. Could you please update the score as you promised？We deeply appreciate your time and efforts.
> > >
> > > Best,
> > >
> > > Authors

---

> > > ### Author Response · Authors · 2023-08-22
> > >
> > > Dear Reviewer Lynn,
> > >
> > > We're very sorry to bother you again, but it's really urgent now because it's almost the deadline for reviewer-author discussion. We are very grateful for your support to our work and willing to improve the score, we sincerely request you to **update the score** in the openreview system.
> > >
> > > Best,
> > >
> > > Authors

---

> ### Author Response · Authors · 2023-08-18
>
> Dear Reviewer Lynm:
>
> We deeply appreciate your constructive feedback, and we have made every effort to address your concerns comprehensively. We kindly ask you to let us know if there are any unresolved concerns. We would welcome the opportunity to engage in further discussions. Thank you for your time.
>
> Best,
>
> Authors.

---

> ### Comment · Area_Chair_Mbwq · 2023-08-20
> **Requesting Feedback and Discussions**
>
> Dear Reviewer Lynn,
>
> I went through your comments, mainly about the technical importance of applying LoRA-like methods to DC. The authors have submitted a response to address this aspect. Could you please check the rebuttal at your earliest convenience?

---

### Official Review · Reviewer_JHfi · 2023-07-14

**Soundness:** 3 good
**Presentation:** 3 good
**Contribution:** 3 good
**Rating:** 6
**Confidence:** 4

**Summary:**

This paper proposes a simple yet effective plugin for data condensation/distillation methods. Specifically, utilizing the low-dimensional nature of image manifolds, it proposes to decompose each synthetic image into two low-rank matrices and optimize them instead. Experimental results demonstrate the effectiveness of the proposed method.

**Strengths:**

- The paper is well-written and easy to follow.

- The idea proposed is novel and interesting.

- The results are promising.

**Weaknesses:**

- There seems to be a gap between the motivation and the method. The motivation is based on the low-dimensionality of natural images while the method is applied to synthetic images. It would be better to include a discussion and experiment on synthetic images to show that they are also intrinsically low-dimensional.

- Missing references and discussions. Although different, the idea proposed is of the same category as the multi-formation in [1] and the Partitioning and Expansion Augmentation scheme in [2]. It would be better to include relevant discussions (preferably with experimental results) in the paper to better position the proposed method in the literature.

[1] Kim, J.H., Kim, J., Oh, S.J., Yun, S., Song, H., Jeong, J., Ha, J.W. and Song, H.O., 2022, June. Dataset condensation via efficient synthetic-data parameterization. In International Conference on Machine Learning (pp. 11102-11118). PMLR.

[2] Zhao, G., Li, G., Qin, Y. and Yu, Y., 2023. Improved distribution matching for dataset condensation. In Proceedings of the IEEE/CVF Conference on Computer Vision and Pattern Recognition (pp. 7856-7865).

**Questions:**

Please address the weaknesses above.

**Limitations:**

The limitations are discussed in the supplementary materials.

---

> ### Author Rebuttal · Authors · 2023-08-10
>
> - **Our motivation and method matching:** As shown in $\underline{\text{Fig.1(a,b) in the response}}$, we perform principal component analysis on real images and synthetic images of DM on the CIFAR10 dataset. It can be observed that both are dominated by a small number of top eigenvalues, so both the real image and the synthetic image of the traditional dataset distillation method(e.g.,DM) are of low rank. Based on this, we propose the dataset distillation in low-rank plugin, which directly performs dataset distillation in the low-rank manifold instead of the original high-dimensional pixel space. The proposed plugin can be applied to arbitrary dataset distillation methods to improve their performance. As shown in the CIFAR10(Img/Cls=1) results in the table below, when our plugin is applied to existing dataset distillation methods, the performance improvement is very significant.
>
> | DC | LoDC(Ours)| DSA | LoDSA(Ours)| DM | LoDM(Ours)| MTT | LoMTT(Ours)|
> | :------:| :------: |:------:| :------: |:------:| :------: |:------:| :------: |
> | 28.3% |  35.2% | 28.8% |  41.0% | 26.0% |  43.8% | 46.3% |  58.7% |
>
>
> - **Related literature discussion:** Thank you for your suggestions for related work. Our motivation and approach are fundamentally different from [1,2], where multi-formation images (in IDC[1]) and partition and expand enhancement schemes (in IDM[2]) exploit the local similarity of image pixels. However, our work exploits the global low-rank properties of images from the perspective of principal component analysis.  Therefore, they are orthogonal, that is, the pixel similarity of the image and the low-rank characteristics of the image are considered at the same time when the dataset is distilled. In our revision, we will discuss these correlations and differences according to your suggestion.  In addition, we also applied our dataset distillation plugin on the latest dataset distillation work (MTT based on gradient trajectory matching). The results on CIFAR10 (Img/Cls=1) are shown below, and our method(LoMTT) shows a clear improvement.
>
> | MTT | IDC[1] | IDM[2] | Ours|
> | :------:| :------: | :------: | :------:
> | 46.3% | 50.6%   |  45.6%  | 58.7% |

---

> > ### Comment · Reviewer_JHfi · 2023-08-18
> > **Thanks for the response**
> >
> > The authors' response addressed most of my concerns and I remain slightly positive about the paper.

---

### Author Rebuttal · Authors · 2023-08-10

Dear reviewers, the figures in the responses are provided in this supplementary material, which contains 3 figures in total:
- Fig.1 is the principal component analysis of real and synthetic images of the dataset distillation method, which shows that both real and synthetic images are low-rank and used to answer the questions of $\underline{\text{reviewers JHfi, Lynm and FpMf}}$.
- Fig.2 is about the distillation performance of the ImageNet dataset at high resolution to answer the questions of $\underline{\text{reviewers Lbw7 and Lynm}}$.
- Fig.3 is about the performance of dataset distillation and continual learning under unbalanced datasets to answer $\underline{\text{reviewer VZYQ}}$'s question.

---

### Decision · Program_Chairs · 2023-09-21

**Decision:**

Accept (poster)

**Comment:**

The authors did a good rebuttal, while the AC agrees with Reviewer FpMf that a thorough analysis and comparison with data parameterization methods will make this work stronger.

The AC would like to support this work, conditioned that a careful revision will be done in the camera-ready